# Analytical yaw models: a two-dimensional comparison

Craig Thompson<sup>1</sup>, Kevin Gouder<sup>1</sup>, and J. Michael Graham<sup>1</sup> Department of Aeronautics, Imperial College London, London, UK

**Correspondence:** Craig Thompson (craig.thompson@imperial.ac.uk)

**Abstract.** Analytical wake models are essential for wind farm design and control. However, they often lack validation beyond scale data. This study compares six, two-dimensional yawed wake models which include single-Gaussian, super-Gaussian, lifting line, and vortex sheet methods. An additional double-Gaussian model is also proposed. All models are calibrated and tested against three datasets: near-wake and far-wake PIV measurements, as well as full-scale turbine data. The proposed double-Gaussian model achieves the lowest mean absolute error (2.6 %) across all datasets. However, all models struggle to predict the full-scale dataset under yawed conditions, emphasising the necessity for validating models against a wide range of turbine operating conditions.

## 1 Introduction

Analytical wake models are mostly used in the developmental stage of wind farms. Their low computational cost makes them particularly useful for the rapid evaluation of farm-scale performance under different turbine locations and operating conditions. Although less detailed than numerical simulations, more recent analytical models are able to accurately predict the entire wake region under large yaw misalignments, thereby increasing their use in the development of farm-wide control algorithms.

Seminal work by Jensen (1983) employs a top-hat shape for the turbine wake which conserves momentum, thereby representing the wake as a uniform velocity deficit expanding linearly downstream. Following from the work by Jensen, it is observed that the shape of the far-wake of a turbine is bell-like, similar to that of a Gaussian profile. The first of the Gaussian-based analytical wake models developed by Bastankhah and Porté-Agel (2014) is validated using small-scale wind tunnel experiments and Large Eddy Simulations (LES) of both small-scale and full-scale turbines. With just a single tuning parameter, the model successfully predicts the far-wake behind a host of turbine geometries subjected to different (non-yawed) inflow conditions. A single-Gaussian profile works well within the far-wake region of a wind turbine, however, the near-wake velocity profile resembles a more complex shape.

Work by Keane et al. (2016) suggests the near-wake shape is that of a double-Gaussian profile. A publication by Schreiber et al. (2020) improved the initial double-Gaussian wake model which has subsequently been further improved by Keane (2021). The final double-Gaussian wake model is validated against LiDAR measurements in the wake of a full-scale 5 MW turbine. The model successfully predicts the full wake of the turbine between distances of x/D = 0.4 to x/D = 5, where D is the rotor diameter. An investigation by Shapiro et al. (2019) introduces a super-Gaussian wake model, consisting of a top-hat shape in the near-wake which morphs to a Gaussian shape in the far-wake. Work by Blondel and Cathelain (2020) proposes an

30

60

alternative form of the super-Gaussian model validated against LES, wind tunnel measurements and full-scale measurements. More recent, three-dimensional models based on the idea of axisymmetric self-similarity assumptions include the introduction of an incoming Atmospheric Boundary Layer (ABL) and wind shear (Ishihara and Qian (2018); Sun and Yang (2018); Li et al. (2022); Tian et al. (2022)).

When developing a wind farm, the largest loss to farm-wide power production occurs when turbines operate directly down-stream of other turbines in their wakes. As a result of the variation in prevailing wind direction, all possible farm layouts result in power loss due to wake interaction at some point in time. To mitigate the loss in power from wake interaction, wake steering is employed. With the upstream turbine purposefully misaligned with the prevailing wind direction, the wake is directed away from the downstream turbine. With the use of a wake steering control system, Medici and Dahlberg (2003) show the possibility of increasing the farm-wide power production. Following from this work, studies by Campagnolo et al. (2016) and Bastankhah and Porté-Agel (2019) experimentally prove, in a scale wind tunnel setting, a wake steering control system can improve the total power output of a set of turbines. Hence, in recent years, wake steering is a focussed area for turbine research.

To model the wake deflection under yawed conditions, work by Bastankhah and Porté-Agel (2016) employs planar Particle Image Velocimetry (PIV) at yaw angles from 0° to 30°. The study proposes a simple single-Gaussian analytical yaw model which successfully predicts the far-wake of a scale turbine at moderate yaw angles. Following this seminal work, Qian and Ishihara (2018) uses eight simulations and an experimental dataset from Bastankhah and Porté-Agel (2016) to tune and validate the coefficients of a single-Gaussian yaw model. Consequently, the proposed analytical yaw model requires no tunable parameters. The single-Gaussian yaw models cannot accurately predict the velocity deficit in the near-wake, but are capable of predicting the deflection of the wake centre, as shown by Bastankhah and Porté-Agel (2016). This results from a linear relation between the wake centre and the downstream distance within the region of the normalised potential core, after which, a more complex relation is necessary.

To model the velocity deficit across the entire wake region of a turbine with yaw misalignment, work by Blondel et al. (2020) adapts their no-yaw super-Gaussian analytical model to include the wake deflection of a turbine under yawed conditions. The method is validated against wind tunnel data from Bastankhah and Porté-Agel (2016). A yawed-3D Jensen-Gaussian full wake model using a double-Gaussian shape is proposed by Zhu et al. (2023) and validated with full-scale LiDAR data of a 1.5 MW turbine. Two recent novel approaches to predicting the wake of a turbine subject to yaw misalignment is a lifting line approach by Shapiro et al. (2018) and a vortex sheet approach by Bastankhah et al. (2022). The model developed by Shapiro et al. (2018) based on lifting line theory treats the yawed turbine as a lifting surface, resulting in more accurate near-disk predictions of transverse velocity and improved subsequent predictions of wake deflection. Similarly to Blondel et al. (2020), the model is validated using wind tunnel data from Bastankhah and Porté-Agel (2016). The vortex sheet model, proposed by Bastankhah et al. (2022) predicts the wake shape by treating the wake edge as a vortex sheet. The model is successful at predicting the wake of turbines subject to yaw misalignment, validated by LES and again by wind tunnel data from Bastankhah and Porté-Agel (2016).

An extensive review of the methods to model wind turbine wakes is given by Amiri et al. (2024) and Wang et al. (2024), these reviews include many analytical models not previously mentioned in this study, however, it is of the authors' opinion that

65

the models mentioned are either the most novel, most accurate, or most validated models available at date of publication. For the evolution of wind turbine aerodynamics, a comprehensive review of turbine flows and the surrounding literature is given by Porté-Agel et al. (2020).

Despite the advancements in analytical wake models, a limitation in the validation of these models is present: the majority of proposed yaw models are validated using a single source of experimental or computational data (often the wind tunnel measurements from Bastankhah and Porté-Agel (2016)). Although this dataset has an extensive FOV at many yaw angles, it is at a much lower Tip Speed Ratio (TSR) and Reynolds number when compared to full-scale turbines. To address these gaps, this paper compares and validates five current analytical yaw models and one proposed analytical yaw model against three datasets. The first dataset is a near-wake high-resolution planar PIV wind tunnel experiment at a working TSR of  $\lambda = 5.7$ , the second is the dataset by Bastankhah and Porté-Agel (2016), and the final dataset is a full-scale 2.2 MW turbine measured in an investigation by Bao et al. (2024).

In the following sections we describe an experimental campaign, a proposed analytical yaw model and the comparison of current yaw models with many experimental datasets. Section 2 includes both the methodology for the planar PIV experiment and the computational methodology to tune the parameters of the analytical models. Following this, Sect. 3 describes a new analytical model based on work by Keane (2021) and Bastankhah and Porté-Agel (2016). The results of the tuning procedure are displayed in Sect. 4 and the resulting model predictions under yaw misalignment are discussed in Sect. 5. Finally, Sect. 6 concludes the findings of this investigation.

#### 2 Methods

## 0 2.1 Experimental methodology

All measurements within this experimental campaign are taken at the  $10^{\circ} \times 5^{\circ}$  wind tunnel located at Imperial College London. The tunnel's test section measures 1.5 m high, 3 m wide, and 20 m long. The free-stream velocity is controlled using a Pitot-static tube placed 1 m upstream of the turbine 1 m from the ceiling in the centre of the tunnel. All measurements within this campaign are taken at a hub-height velocity of 7.8 ms<sup>-1</sup> with a turbulence intensity of 5.3 %. Thirteen yaw angles between  $\pm 30^{\circ}$  in steps of  $5^{\circ}$  are measured.

An ABL is required for this experimental campaign. Therefore, an initial investigation is conducted to ensure the correct boundary layer is developed.

### 2.1.1 Atmospheric boundary layer

The ABL chosen for the scale experiments is that experienced by offshore farms, more commonly called a "marine" boundary layer. Work by Irwin (1981) states a design procedure for static upstream spires to produce a scaled ABL comparable, for example, to that experienced by offshore farms. Figure 1 presents the normalised mean velocity and turbulence intensity profiles from a preliminary investigation of a two-spire configuration, measured using Laser Doppler Anemometry (LDA), and

Figure 1. Mean normalised streamwise (Left) velocity and (Right) intensity compared with ISO 19901-1:2015.

compares them with ISO 19901-1:2015 profiles. The investigation concludes that spires of 1.5 m high, 0.166 m at the base, 0.073 m at the top, and spaced 1.3 m apart produce an ABL consistent with that outlined in the ISO 19901-1:2015. Hence, this configuration is chosen for the experimental campaign.

## 2.1.2 Turbine design

100

A 1:250 scale, three-bladed turbine with a 0.5 m diameter and 0.38 m hub height is designed based on the Senvion 6M126 offshore turbine. The turbine's blades are developed such that the power coefficient ( $C_p = 0.35$ ) and TSR ( $\lambda = 5.7$ ) of the model turbine are within 15 % of the full scale equivalent. More information on the blade development can be found in Appendix A.

To ensure the TSR is kept constant, a Maxon DC 200W RE50 motor operating in a generator configuration controlled by a Maxon ESCON 50/5 servo controller, which reads rotational velocity using an optical encoder are employed. The servo controller allows both current and rotational velocity control, adjusted through the linear relationship with an analogue input value generated by an NI USB-6229 and controlled via LabVIEW.

The turbine is equipped with a DYN-205 rotary torque sensor, rated for 0.5 Nm, and a SMD S251 load cell, rated for 10 N. These devices measure the torque and axial thrust respectively. Measurements are taken via an NI USB-6229 controlled using LabVIEW at a frequency of 1 kHz.

#### 2.1.3 Wake measurements

Figure 2 shows the planar PIV set-up used to measure the wake of the wind turbine. Four high-speed Phantom V641 cameras are mounted above the ceiling of the test section. The cameras are fitted with Nikon AF Nikkor 50 mm f/1.4D lenses set to an aperture of f/1.8. A high-speed Litron LDY304 laser, equipped with a Dantec laser sheet optic, provides a 2 mm thick light sheet

**Figure 2.** (Left) Technical diagram and (Right) image of the near-wake wind tunnel experiments. The dashed lines show each cameras FOV and the black triangles represent the spires.

with a  $36.8^{\circ}$  divergence angle, parallel to the tunnel floor at the turbine hub height of 0.38 m. The sheet illuminates polyethylene glycol (PEG) seeding pumped into the test section. The centre of the PIV field is located 2.7D (1.35 m) downstream from the centre of the turbines blades at a yaw angle of  $0^{\circ}$ , with a FOV of 2D (1 m) in the spanwise direction and 1.2D (0.6 m) in the streamwise direction.

The cameras and laser are controlled by the PIV system developed by LaVision, which is implemented in the software DaVis 10. The software manages the data acquisition, storage and timing of the triggers to the laser and cameras. A desktop computer with an external programmable timing unit manufactured by LaVision is used to ensure the relative timing of these signals. A non time-resolved dataset, consisting of 3000 snapshots taken at a frequency of less than 10 Hz is taken for each yaw angle. All turbine measurements (torque, angular velocity and thrust) are taken at 1 kHz for the entire duration of the planar PIV measurements. To ensure the velocity snapshots are synchronous with the turbine measurements, a trigger signal used to initiate the PIV measurements is connected to a channel of the turbine data acquisition.

Figure 3. Mean of the streamwise velocity in the wake of the turbine at zero degrees yaw angle.

Initially, three filters are applied to the images: a minimum subtraction in time is first used to remove the smallest intensity captured by a pixel in time that can be thought of as the image's background. A local minimum subtraction is used next, with a window size equivalent to the final interrogation window, and can be thought of as local noise removal. Finally, a min-max filter, which can be interpreted as a binary filter, is then applied.

After the raw images have been pre-processed, a multi-pass cross-correlation procedure is employed using DaVis 10. Interrogation windows of size  $96 \text{ px} \times 96 \text{ px}$ ,  $48 \text{ px} \times 48 \text{ px}$  and  $24 \text{ px} \times 24 \text{ px}$  are used for the cross-correlation with an overlap of 75 %. The four vector fields are then stitched together to produce a final vector field consisting of one vector every 1.847 mm. Figure 3 displays the mean streamwise velocity in the wake of the turbine at zero degrees yaw angle.

# 2.2 Computational methodology

130

The models compared within this investigation, along with some of their key information are displayed in Table 1. To ensure a consistent and unbiased analysis of the wake models under yaw misalignment, each model's parameters are tuned at  $\gamma = 0^{\circ}$ ,

**Table 1.** Analytical models used within this investigation.

| Model                            | Acronym | Method                       | Tunable parameters |
|----------------------------------|---------|------------------------------|--------------------|
| Bastankhah and Porté-Agel (2016) | BPA     | Single-Gaussian              | 3                  |
| Shapiro et al. (2018)            | Sh      | Lifting line single-Gaussian | 2                  |
| Qian and Ishihara (2018)         | QI      | Single-Gaussian              | 0                  |
| Blondel et al. (2020)            | Bl      | Super-Gaussian               | 3                  |
| Bastankhah et al. (2022)         | Ba      | Vortex sheet single-Gaussian | 2                  |
| Present                          | Pr      | Double-Gaussian              | 4                  |

with the exception of the model by Qian and Ishihara (2018) as no tuning parameters are required. The model's parameters are tuned by minimising the Mean Absolute Error (MAE) shown below

$$MAE = \frac{1}{N} \sum_{i=1}^{N} |U_{i,exp} - U_{i,pred}|,$$
(1)

where N is the number of data-points,  $U_{i,exp}$  is the experimental velocity and  $U_{i,pred}$  is the model predicted velocity. The minimisation is performed with the MATLAB function fminsearch, where the initial guess x0 corresponds to the value used in the model validation of the relevant publication.

### 140 3 Proposed model

This section proposes a new wake model, developed to improve the accuracy of predictions in the near-wake under yaw misalignment. The proposed wake model inherits a double Gaussian shape from work by Keane (2021) and a wake centre  $y_c$  from work by Bastankhah and Porté-Agel (2016).

The final velocity deficit is in the form

145 
$$\frac{\Delta U}{U_{\infty}} = C(x)f(r,\sigma(x),\gamma),$$
 (2)

where  $U_{\infty}$  is the incoming flow velocity, the wake decay function C(x) is defined by the conservation of momentum using the actuator disc model and the wake shape function  $f(r, \sigma(x), \gamma)$  determines the double-Gaussian shape given by

$$f(r,\sigma(x),\gamma) = \frac{1}{2} [exp(D_{+}) + exp(D_{-})], \tag{3}$$

where  $D_{\pm}$  is given by

150 
$$D_{\pm} = -\frac{1}{2}\sigma(x)^{-2}((r+y_c)\pm\cos(\gamma)^2r_{min})^2$$
. (4)

The minimum radius for the double-Gaussian displacement  $r_{min}$  is determined from the tunable parameter  $r'_{min} = r_{min}/R$ , where R is the radius of the turbine. The single Gaussian function  $\sigma$  is given as

$$\sigma = \alpha x^n + \epsilon D,\tag{5}$$

where  $\epsilon = (d'_e - r'_{min})/6$ . The final three tunable parameters are therefore  $d'_e$ ,  $\alpha$  and n. The equation for the wake centre  $y_c$  is dependant on the normalised length of the potential core  $x_0/D$  given below

$$\frac{x_0}{D} = \frac{\cos(\gamma)(1 + \sqrt{1 - C_T})}{\sqrt{2}(2.32T_i + 0.154(1 - \sqrt{1 - C_T})},\tag{6}$$

where  $C_T$  is the axial thrust coefficient and  $T_i$  is the incoming turbulence intensity.

Close to the rotor, the displacement of the wake centre increases linearly with the downstream distance such that

$$x \le x_0 : \frac{y_c}{D} = \theta \frac{x}{D},\tag{7}$$

where the wake skew angle  $\theta$  is given as

$$\theta = \frac{0.3\gamma}{\cos(\gamma)} \left( 1 - \sqrt{1 - C_T \cos(\gamma)} \right). \tag{8}$$

Further from the rotor, the displacement of the wake centre is determined as followed

$$x > x_0 : \frac{y_c}{D} = \theta \frac{x_0}{D} + \frac{\theta}{14.7} \sqrt{\frac{\cos(\gamma)}{k^2 C_T}} (2.9 + 1.3\sqrt{1 - C_T} - C_T) ln \left[ \frac{(1.6 + \sqrt{C_T}) \left( 1.6\sqrt{\frac{\bar{\sigma}^2}{D^2}} \frac{8}{\cos(\gamma)} - \sqrt{C_T} \right)}{(1.6 - \sqrt{C_T}) \left( 1.6\sqrt{\frac{\bar{\sigma}^2}{D^2}} \frac{8}{\cos(\gamma)} + \sqrt{C_T} \right)} \right], \tag{9}$$

to determine the wake centre  $y_c$  an additional definition of the square of the wake widths  $\bar{\sigma}^2$  is used, given as

165 
$$\frac{\bar{\sigma}^2}{D^2} = \left(k\frac{x - x_0}{D} + \frac{\cos(\gamma)}{\sqrt{8}}\right) \left(k\frac{x - x_0}{D} + \frac{1}{\sqrt{8}}\right).$$
 (10)

The wake centre  $y_c$  is experimentally shown by Bastankhah and Porté-Agel (2016) to remain at the hub height for yaw angles less than  $\pm 30^{\circ}$ , hence, the limitations of this two-dimensional model is  $\gamma \leq \pm 30^{\circ}$ .

The wake growth rate has the empirical relation suggested by Qian and Ishihara (2018) shown below

$$k = 0.11C_T^{1.07}T_i^{0.2}. (11)$$

With the double-Gaussian shape function  $f(r, \sigma(x), \gamma)$  specified, the wake decay function C(x) remains to be determined. From the actuator disc model, Tennekes and Lumley (1972) show the mean momentum flux across a disk is given by

$$\rho \int U(U_{\infty} - U)dA = T,\tag{12}$$

where  $\rho$  is the fluid density, A is the cross-sectional area of the actuator disk and T is the total force. The total force is considered to be the effective thrust, shown below

175 
$$T = \frac{1}{2} C_T \rho A_e U_\infty^2$$
, (13)

where  $A_e$  is the effective rotor area, such that  $A_e=\pi d_e^2/4$ , where  $d_e$  is the effective rotor diameter determined from the tunable parameter  $d_e'=d_e/D$ . Combining Eq. 12 and Eq. 13, the integral below

$$\int U(U_{\infty} - U)dA = \frac{1}{2}C_T A_e U_{\infty}^2,\tag{14}$$

has a quadratic solution in the form of

180 
$$NC^2(x) - MC(x) + \frac{1}{8}C_T d_e^2 = 0,$$
 (15)

$$N = \sigma^2 exp(-\sigma^{-2}r_{min}^2) + \frac{1}{2}\sqrt{\pi}r_{min}\sigma erf(\sigma^{-1}r_{min}), \tag{16}$$

$$M = 2\sigma^2 exp(-\frac{1}{2}\sigma^{-2}r_{min}^2) + \sqrt{2\pi}r_{min}\sigma erf(\frac{\sigma^{-1}r_{min}}{\sqrt{2}}),\tag{17}$$

where erf is the error function. The discriminant of Eq. 15 is

$$S = M^2 - \frac{1}{2}NC_T d_e^2. {18}$$

The roots of Eq. 15 and therefore the solution to the decay function C(x) is dependant on the sign of the discriminant S, such that

$$S \ge 0: C(x) = \frac{M - S^{\frac{1}{2}}}{2N},\tag{19}$$

$$S < 0: C(x) = \frac{(M^2 + |S|)^{\frac{1}{2}}}{2N}.$$
 (20)

## 3.1 Model procedure

A complete form of the procedure for predicting the wake of a turbine under yaw using the proposed model is as follows:

- 1. Calculate the single Gaussian function  $\sigma$  using Eq. 5, the normalised length of the potential core  $x_0/D$  using Eq. 6, the wake skew angle  $\theta$  using Eq. 8 and the wake growth rate k using Eq. 11.
- 2. Determine the square of the wake widths  $\bar{\sigma}^2$  using Eq. 10.
- 3. Calculate the normalised wake centre  $y_c/D$  using Eq. 7 for when  $x \le x_0$  and Eq. 9 for when  $x > x_0$ .
- 4. Determine the wake shape function f in Eq. 3 by first calculating  $D_{\pm}$  using Eq. 4.
- 5. Calculate N, M and then S using Eq. 16, Eq. 17 and Eq. 18, respectively.
- 6. For  $S \ge 0$  use Eq. 19 to find the wake decay function C and for S 

230

Figure 4. Spanwise profiles of the normalised streamwise velocity at x/D=1.05, 1.65 and 3, comparing near-wake experimental data with tuned models at a yaw angle of zero degrees. The solid vertical grey lines indicate  $U/U_{\infty}=0.5$ .

reasonable accuracy, with the exception of the single-Gaussian model proposed by Qian and Ishihara (2018), which underpredicts the magnitude of the deficit across all downstream locations. The vortex sheet model performs well for most streamwise positions aside from a slight underprediction of the velocity deficit at x/D=4. The double-Gaussian model provides the most accurate representation of the wake profile edges, particularly at the downstream locations.

Figure 6 presents the analytical wake models at three streamwise positions, tuned using the full-scale data of Bao et al. (2024). The single-Gaussian model of Qian and Ishihara (2018) and the vortex sheet model exhibit similar behaviour, overpredicting the velocity deficit in the near-wake region close to the turbine before transitioning to an underprediction further downstream. All models underpredict the wake deficit at the edges close to the turbine and then overpredict the deficit further downstream.

Figure 7 presents the MAE and Root Mean Square Error (RMSE) for each of the models tuned on the three datasets. In the near-wake region, the double-Gaussian and super-Gaussian models perform far better than the single-Gaussian models with a MAE of less than 1.5 %. For the far-wake dataset, all models achieve a MAE within 2.5 % of the experimental data, with the exception of the single-Gaussian model proposed by Qian and Ishihara (2018). In the full-scale case, all models exhibit a MAE below 4 %. As shown by the tuned velocity profiles, the super-Gaussian and double-Gaussian models provide the best overall performance across all datasets with a maximum MAE of 1.6 %. The reasoning for the greater accuracy of the methods is a consequence of their more complex representation of the wake profile, alongside the greater number of tuning parameters, thereby allowing the models greater flexibility for differing turbine operating conditions. In contrast, the single-Gaussian model proposed by Qian and Ishihara (2018) consistently underperformed, with a maximum MAE of 10.5 % for the

Figure 5. Spanwise profiles of the normalised streamwise velocity from x/D=4 to 11, comparing far-wake experimental data with tuned models at a yaw angle of zero degrees. The solid vertical grey lines indicate  $U/U_{\infty}=0.25$ .

Figure 6. Spanwise profiles of the normalised streamwise velocity at x/D=2, 4 and 6, comparing full-scale experimental data with tuned models at a yaw angle of zero degrees. The solid vertical grey lines indicate  $U/U_{\infty}=0.25$ .

near-wake dataset. The single-Gaussian model has no tunable parameters, hence is solely relying on input parameters, which limits its ability to accurately capture variations in the wake structure across different operating conditions.

**Figure 7.** MAE and RMSE, expressed as percentages of the free-stream velocity, for each analytical model tuned against the near-wake, far-wake, and full-scale velocity data.

#### 5 Model predictions

After the tuning parameters are determined, the analytical models can be used to predict the wake behind turbines subject to yaw misalignment. The following section discusses the ability of each model to accurately predict the near-wake scale data at yaw angles ranging from  $\pm 30^{\circ}$ , the far-wake scale data from work by Bastankhah and Porté-Agel (2016) at yaw angles of 10 and  $20^{\circ}$ , and the full-scale data from work by Bao et al. (2024) at a yaw angle of  $11^{\circ}$ .

Figure 8 presents the wake prediction from the analytical models at three streamwise positions for yaw angles of  $10^{\circ}$  and  $30^{\circ}$  against the near-wake PIV dataset. The super-Gaussian model consistently underpredicts the velocity deficit at both yaw angles. Similarly, especially downstream, the single-Gaussian model proposed by Qian and Ishihara (2018) undepredicts the velocity deficit at a yaw angle of  $10^{\circ}$ . In contrast, the vortex sheet model overestimates the deficit in the near-wake before slightly underpredicting it further downstream for both angles, again, indicating a mismatch in the wake recovery observed when tuning the model. Aside from the lifting line model and the double-Gaussian model, all models underpredict the velocity deficit at the downstream location.

In terms of the prediction of the wake centre, all models capture the deflection well, other than the super-Gaussian model which underpredicts the wake centre at a 30° yaw angle. Both the lifting line model and the double-Gaussian model maintain good agreement with the measured profiles across all positions and yaw angles, effectively capturing both the velocity deficit and the wake centre throughout the near-wake region. At large yaw angles it is observed that the near-wake profile becomes more Gaussian, hence why the single-Gaussian lifting line model performs accurately. This phenomenon is captured within Eq. 4 with the multiplication of  $r_{min}$  by  $cos(\gamma)^2$ .

Figure 9 presents the MAE and RMSE of the analytical models against the experimental PIV data for yaw angles ranging from  $-30^{\circ}$  to  $30^{\circ}$  in the near-wake region. The single-Gaussian models, particularly the lifting line formulation, perform best at large yaw angles, which is likely due to the wake profile becoming more Gaussian and less top-hat in shape under extreme misalignment. For angles below  $20^{\circ}$  the single-Gaussian model developed by Qian and Ishihara (2018) exhibits the largest

265

Figure 8. Spanwise profiles of normalised streamwise velocity at  $x/D=1.05,\,1.65,\,$  and 3, comparing near-wake experimental data with tuned models at (Top)  $\gamma=10^{\circ}$ , and (Bottom)  $\gamma=30^{\circ}$ . The solid vertical grey line marks  $U/U_{\infty}=0.5$ .

errors, whereas the remaining models maintain MAE values around 5 %. The super-Gaussian model performs poorly at large yaw angles, while the double-Gaussian model consistently exhibits MAE values below 2.5 % across all yaw cases.

Figure 10 shows the wake prediction from the analytical models at streamwise locations from x/D=4 to x/D=11 for yaw angles of  $10^{\circ}$  and  $20^{\circ}$  compared to the far-wake scale data by Bastankhah and Porté-Agel (2016). At a yaw angle of  $10^{\circ}$ , all models reproduce the wake deficit well, with the exception of the single-Gaussian model proposed by Qian and Ishihara (2018), which underpredicts the velocity deficit at both yaw angles. At a yaw angle of  $20^{\circ}$ , aside from at x/D=4, all yaw models slightly overpredict the wake deficit.

Similarly to the near-wake results, the super-Gaussian model underpredicts the yawed wake position at  $10^{\circ}$ . This underprediction becomes more pronounced at a yaw angle of  $20^{\circ}$ . In contrast, the single-Gaussian model by Qian and Ishihara (2018) is seen to overpredict the wake centre at  $20^{\circ}$ . All other models generally maintain good agreement with the measured wake centre, although a slight overprediction is observed further downstream at  $20^{\circ}$ .

Figure 11 presents the velocity deficit of the analytical models at 2D, 4D and 6D downstream at a yaw angle of  $11^{\circ}$  compared to the full-scale dataset by Bao et al. (2024). At 2D, the single-Gaussian model developed by Qian and Ishihara (2018) provides the closest agreement with the measured velocity deficit, while the vortex sheet model overpredicts the deficit

Figure 9. MAE and RMSE, expressed as percentages of the free-stream velocity, for each analytical model compared with near-wake velocity data at yaw angles ranging from  $\gamma = -30^{\circ}$  to  $\gamma = 30^{\circ}$ 

and all other models underpredict it. Further downstream, all models perform similarly, exhibiting a noticeable underprediction of the velocity deficit, highlighting an inaccuracy of current analytical yaw models being developed using only numerical or scale experimental data.

At 2D, the model's wake deflection predictions slightly underpredict the wake centre, however they generally agree with the measurements. This misprediction is exaggerated further downstream at 4D, where all models largely underpredict the yawed wake position. However, at 6D the misalignment of the wake centre reduces, showing a more reasonable match to the full-scale data. The super-Gaussian model exhibits a marginally greater underprediction of the wake centre than the other models, consistent with trends observed in the near and far-wake discussion.

Figure 12 compares the performance of all analytical models across the three experimental datasets at yaw angles of  $10^{\circ}$  for the near and far-wake scale datasets and  $11^{\circ}$  for the full-scale dataset, together with their mean performance. In the near-wake, the single-Gaussian model developed by Qian and Ishihara (2018) performs the worst, while the single-Gaussian model by Bastankhah and Porté-Agel (2016), the vortex sheet model, and lifting line model show similar behaviour, with MAE values between 5.2% and 6.2%. Both the super-Gaussian and double-Gaussian models closely match the experimental data, with the double-Gaussian model achieving a MAE of just 2.2%. The far-wake dataset exhibits similar trends, with the single-Gaussian model by Qian and Ishihara (2018) consistently yielding the highest errors ( $\approx 5\%$ ) and the remaining models maintaining MAE values below 2.5%. The double-Gaussian model again performs best, achieving a MAE of 1.5%.

Figure 10. Spanwise profiles of normalised streamwise velocity from x/D=4 to 11, comparing far-wake experimental data with tuned models at (Top)  $\gamma=10^{\circ}$ , and (Bottom)  $\gamma=20^{\circ}$ . The solid vertical grey line marks  $U/U_{\infty}=0.25$ .

For the full-scale case all models display comparable predictive accuracy, with a MAE within  $\pm 0.7$  %. The super-Gaussian model performed the worst, resulting from both greater yaw missalignment and underprediction of the wake centre. When averaging across all datasets, it is evident that the single-Gaussian model by Qian and Ishihara (2018) has the poorest predictive capability, as expected given its lack of tunable parameters. The super-Gaussian, single-Gaussian model by Bastankhah and Porté-Agel (2016), lifting line, and vortex sheet methods exhibit similar performance, with MAE values ranging between 3.6 % and 4.2 %. The double-Gaussian model consistently demonstrates the highest accuracy, achieving an average MAE of 2.6 %, akin to the model's flexibility resulting from having four tunable parameters.

#### 295 6 Conclusions

This study compared six analytical yaw wake models, including a newly proposed double-Gaussian formulation based on the work of Keane (2021) and Bastankhah and Porté-Agel (2016), against three datasets: a near-wake scale PIV experiment, the far-wake scale data of Bastankhah and Porté-Agel (2016), and the full-scale measurements of Bao et al. (2024). All models are first tuned at zero yaw to ensure consistent baseline performance before being evaluated under yaw misalignments of up to  $\pm 30^{\circ}$ .

Figure 11. Spanwise profiles of normalised streamwise velocity at x/D=2, 4 and 6, comparing full-scale experimental data with tuned models at  $\gamma=11^{\circ}$ . The solid vertical grey line marks  $U/U_{\infty}=0.25$ .

Figure 12. MAE and RMSE, expressed as percentages of the free-stream velocity, for each analytical model compared with near-wake  $(\gamma = 10^{\circ})$ , far-wake  $(\gamma = 10^{\circ})$ , and full-scale  $(\gamma = 11^{\circ})$  velocity data, along with the mean of all three datasets.

From the near-wake comparison, the super-Gaussian and double-Gaussian models perform best, reinforcing that a more complex shape than the single-Gaussian is required, particularly at small yaw angles. The far-wake comparison showed good predictions from most models, except for the single-Gaussian model by Qian and Ishihara (2018), with the double-Gaussian model providing marginally more accurate results. Finally, the full-scale comparison indicates that all the models are inaccurate in predicting both the wake deficit and centre in the downstream region for a full-scale turbine under yawed conditions.

Overall the proposed double-Gaussian model outperformed all other models for both near and far-wake predictions. However it too struggled to correctly predict the full-scale data. This complete inaccuracy highlights the importance of validating models against a wide range of turbine operating conditions. Hence, there is a need for more full-scale turbine datasets in which measurements are taken in the wake of turbines under yaw misalignment.

Data availability. All data supporting this study are openly available from Imperial College London's repository at [ADD THIS IN]

### Appendix A: Blade development

The turbine blades are developed using the Blade Element Momentum (BEM) method, where chord distribution is calculated based on the Betz optimum:

$$c = S_f \frac{8\pi r}{BC_I} (1 - \cos(\varphi)),\tag{A1}$$

where  $S_f$  is a scaling factor, r is the section radius, B is the number of blades,  $C_l$  is the sectional lift coefficient at maximum lift-to-drag ratio, and  $\varphi$  is the local flow angle. The local flow angle  $\varphi$  is determined by

$$\varphi = \frac{2}{3} \tan^{-1} \left( \frac{1}{\lambda_i} \right), \tag{A2}$$

where  $\lambda_i$  is the sectional TSR. Based on the optimised chord distribution, the L-BFGS algorithm is implemented to maximise the power of each blade section (calculated from the BEM method) by modifying the twist of that section. The low Reynolds number SG6043 airfoil Giguere and Selig (1998) is blended along the blade at 20 discrete spanwise positions from the 0.05 m diameter hub. Table A1 shows the normalised chord and twist distribution of the blade at each position for  $S_f = 1.4$ .

**A1** 

Author contributions. CT carried out measurements, analysed the data and wrote the first draft. KG and MG supervised the work, edited several drafts, secured funding and managed the project.

Competing interests. The authors declare no conflict of interest.

Table A1. Spanwise blade distribution.

| b/R [-]  | Airfoil [-] | c/R [-]  | Twist [deg] |
|----------|-------------|----------|-------------|
| 0.1      | Circle      | 0.08     | 0           |
| 0.147368 | SG6043      | 0.241164 | 31.9798     |
| 0.194737 | SG6043      | 0.234241 | 26.7401     |
| 0.242105 | SG6043      | 0.218919 | 22.4622     |
| 0.289474 | SG6043      | 0.201538 | 19.2407     |
| 0.336842 | SG6043      | 0.184726 | 16.6767     |
| 0.384211 | SG6043      | 0.169416 | 14.5713     |
| 0.431579 | SG6043      | 0.155817 | 12.8164     |
| 0.478947 | SG6043      | 0.143853 | 11.3495     |
| 0.526316 | SG6043      | 0.133349 | 10.1085     |
| 0.573684 | SG6043      | 0.124112 | 9.0468      |
| 0.621053 | SG6043      | 0.115962 | 8.13403     |
| 0.668421 | SG6043      | 0.108738 | 7.34268     |
| 0.715789 | SG6043      | 0.102307 | 6.6759      |
| 0.763158 | SG6043      | 0.096554 | 6.1324      |
| 0.810526 | SG6043      | 0.091384 | 5.75058     |
| 0.857895 | SG6043      | 0.086717 | 5.31337     |
| 0.905263 | SG6043      | 0.082486 | 5.0468      |
| 0.952632 | SG6043      | 0.078636 | 4.78023     |
| 1        | SG6043      | 0.075969 | 4.73474     |
|          |             |          |             |

Acknowledgements. We are indebted to EPSRC through grant EP/L024888/1 for access to the National Wind Tunnel Facility (NWTF). We are very grateful for funding by the European Commission for project ICONIC, through which the turbines and wind tunnel time were funded. The authors gratefully acknowledge the support of colleagues in the Department of Aeronautics at Imperial College London, including Mr Will McArdle, Mr Ricardo Huerta Cruz, Mr Paul Howard, Mr Mark Grant, and Ms Dominica Rohozinski, among others. The authors acknowledge the use of LLMs for grammatical consistency.

19

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
