# Peer review of "Analytical yaw models: a two-dimensional comparison"

_Wind Energy Science, 2025_

## Referee Comment (RC2)

**Comments to the Authors**
**Paper title: Analytical yaw models: a two-dimensional comparison**

**1 General comments**

This paper presents a novel set of measurements characterising the near wake of a yawed, small-scale wind turbine in a wind tunnel. By combining these measurements with existing literature data, the authors compare several analytical velocity deficit models coupled with yaw models. A model is proposed and evaluated alongside established approaches. Before comparison, all models are calibrated using non-yawed wind turbine wake data, ensuring a consistent methodology across the analysis. The proposed paper is interesting and includes a new dataset for model validation. However, this first submission suffers from limitations that will be detailed later. Also, the potential impact of this study and its innovative character are not clear to me.

**2 Specific comments**

- The analysis is restricted to lateral profiles, which may be misleading. Wake centers are not consistently located near hub height, contrary to the paper's assumptions. Expanding measurements to include 2D $y - z$ planes would provide a more comprehensive understanding of wake behaviour.

- The study's focus on the near wake is limiting in view of validating models. Extending measurements to cover the full wake would offer deeper insights into wake evolution and model applicability.

- The calibration process is unclear and requires significant clarification. Specifically:

  - Which parameters are calibrated, and how?
  - Do parameters vary with streamwise location, or are predefined functions assumed (e.g., a linear approach for the Gaussian width $\sigma$ or another function for the super-Gaussian order $n$)?
  - If parameters are set for each streamwise location, how are integrations performed with the super-Gaussian model?

- Was the presented model implementation validated in their original form, before calibration?

- The presented results for the super-Gaussian model deviate from those in other studies using the same dataset (e.g., wind tunnel data from Bastankhah & Porté-Agel). This discrepancy must be addressed, possibly by validating the implementation. Additionally, reporting calibrated values and comparing to accepted model calibration (i.e., https://doi.org/10.5194/wes-8-141-2023 in the present case) would provide some insight. A sensitivity study on the calibration constant could also provide insight.

- Missing information to reproduce the results: based on the information provided in the paper, it is not possible to reproduce the test cases. At least the wind turbine thrust coefficients must be provided, together with other relevant flow and wind turbine operation variables.

- The method used to estimate the yawed thrust coefficient ($C_{T,\Psi}$ with $\Psi$ the yaw angle) is unclear. Different approaches exist (e.g., $C_T cos(\Psi)$, $C_T cos^2(\Psi)$, $C_T cos^3(\Psi)$), and consistency across models is essential for fair comparison. A clear presentation of all models and their assumptions is needed.

- The "new" approach combines two existing models, which raises questions about its novelty and the space devoted to it in the paper. If something new is introduced here, please mention it clearly.

- While mean streamwise velocities and turbulence intensity are compared to ISO standards, further details are required:

- Clear definition of turbulence intensity (TI)

- Assessment of the lateral velocity component

- Uniformity of velocity profiles in the lateral direction

- Comparison of integral length scales to expectations

- Provision of both lateral and vertical profiles

- Beyond the new dataset, the study's novelty is unclear. Recent advancements in modelling complex wake shapes under yaw and secondary steering are not addressed, limiting the paper's contribution to the field.

- Surprisingly, the widely used Jimenez model is omitted from the comparison. Including it would provide a more comprehensive benchmark for the proposed and existing models.

**3   Technical corrections**

1. L.2: The use of "2D" is ambiguous. Please clarify.

2. L.3: The claim that a "new" double-Gaussian model is proposed is misleading. The model used is that of Keane et al., coupled with the Bastankhah & Porté-Agel yaw model. Please clarify what is new in the proposed approach.

3. L.5: The assertion that measurements serve as an undebatable reference overlooks the complexity of full-scale measurements, which are subject to uncontrolled environmental biases. Are these measurements based on neutral atmospheric conditions? A more critical discussion is needed.

4. L.11: The term "more" is vague. Please reformulate for precision and provide references to support the claim.

5. L.17: While the model depends on a single tuning parameter ($\sigma$, I assume), $\sigma$ itself is usually assumed to be a linear function with coefficients dependent on turbine operating and environmental conditions. This should be made explicit.

6. L.29: The mention of wind shear in the Ishihara & Qian paper is unclear. Shear is not a parameter in the models, and previous models (e.g., Bastankhah & Porté-Agel) are compatible with non-uniform vertical streamwise velocity profiles as input. Clarify the intended meaning.

7. L.37: Insert "that" between "setting" and "a wake steering" for grammatical correctness.

8. L.43: The statement "requires no tunable parameters" is misleading, as the model depends on multiple calibration constants. Furthermore, these constants could also be tuned in this study.

9. A discussion on wake deformation (curled-wake effect) and the complex wake shape behind yawed turbines should be included in the introduction.

10. L.81: please use the international system of units

11. L.84: Define turbulence intensity (TI) clearly: is it based on hub-height velocity or local velocity u(z)?

12. L.85: Both positive and negative yaw angles were considered. Did you observe any non-symmetric behavior as claimed in some studies (https://doi.org/10.5194/wes-6-1521-2021)? This should be discussed.

13. Figure 1.: Define TI and include $V/U_h$, $Ti_v$, $Ti_w$, and turbulent length scales for completeness.

14. Section 2.1.2: Please provide thrust coefficients for both unyawed and yawed cases. Is the current controller representative of real-scale turbines, especially regarding yaw misalignment? L.119: Did you verify the statistical convergence of the procedure? L.123 to L.126: Are these filtering operations standard? If so, provide references.

15. What is the main motivation for focusing on near-wake characteristics in these measurements?

16. Section 2.2: The calibration process is unclear. Specify which parameters are tuned, and whether they are tuned independently for each streamwise location and test case. Explain how integration is performed in the super-Gaussian model if the parameters are not continuous.

17. Section 2.2: It would be informative to compare tuned models to their standard calibration, possibly in an appendix.

18. Section 3: The derivation appears to use the double-Gaussian formulation of Keane et al. with the Bastankhah & Porté-Agel yaw model. If no new models are proposed, the extensive derivation should be justified or condensed. If new elements are introduced, they should be clearly highlighted.

19. Eq.2: Explicitly state that $\gamma$ corresponds to the yaw angle. Ensure all quantities in the derivation are clearly defined.

20. L.152: $\sigma$ is not a Gaussian function; please correct this phrasing.

21. Eq.6: Provide a reference to the appropriate source for this model.

22. L.158: Please clearly distinguish between non-yawed and yawed $C_T$ throughout the paper.

23. L.166: The claim that the wake center is aligned with hub-height is invalid. Bastankhah & Porté-Agel (Figure 5) clearly shows otherwise.

24. Figure 4.: The x-axis should not be the streamwise distance, $x/D$, but rather $x/D + (u - u_h)/u_h$ (or similar).

25. Figure 5. he poor agreement between Ishihara & Qian's model and measurements is surprising. Was the implementation validated against the original paper's test cases?

26. L.240: the experimental data also contain a test case at $30^o$. Why was this not considered in the analysis?

27. L.253: The use of a square cosine function to modify $r_{min}$ requires justification. A simple cosine might be more appropriate for a purely trigonometric transformation.

---

## Author Comment (AC1)

**Referee:1**

General comments:

The paper compares six analytical wake models with three different experimental datasets

for the near-wake (x/D=2 – 3.5) of a model-scale turbine, for the far-wake (x/D= 4-11) of another model-scale turbine, and for the intermediate wake (x/D=2-6) of a full-scale wind turbine. The model comparisons are first tuned and then compared for a non-yawed turbine wake and thereafter compared for several yawed wake configurations.

The work includes a novel experimental dataset (near wake model scale) and two existing datasets from other publications. Furthermore, the work proposes a new wake model with yaw-capabilities. From the model description it seems that the new model mainly combines the wake deflection sub-model by Bastankhah and Porte-Agel (2016) and the double Gaussian deficit model by Keane (2021). The model description should be clearer about which equations were adapted from these models, and which are new additions.

The paper is well-organized, features concise language, and is very interesting to read. It features a detailed analysis of the modeled mean velocity deficits and a quantification of the mean absolute error for the different wake configurations.

The paper addresses the scientifically relevant topic of improved analytical wake modelling for non-yawed and yawed cases, while specifically the near-wake modelling and the full-scale applicability are highlighted as novel contributions. Full-scale wake flow validations are indeed considered essential, and the authors emphasize the need for validating yawed wake models for further full-scale cases. The need for accurate modelling of the near-wake could be justified more, as it is not completely clear how wind farm models would benefit from improved near- wake modeling. The importance of accurate near-wake models for wind farm simulations should be motivated in more detail.

We thank the reviewer for providing valuable positive feedback on our manuscript. In the following document we address each of their comments, enhancing the revised version to meet the standards for publication. Any modifications made to the manuscript are highlighted in blue and, when relevant, have been included below to address the reviewer's specific comment.

The authors would like to take a moment to highlight the novelty of this submission, which comes in three forms:

- The inclusion of a high resolution near-wake dataset.

- A comparison of current analytical models using three experimental datasets (including full-scale), where previous literature would have mostly only included one experimental dataset.

- The proposition of an analytical model as the combination of two previous models.

The authors are thankful to the reviewer for their valuable insight on addressing motivation of the importance of accurate near-wake models. It is of the authors opinion that the near-wake modelling is necessary because:

- Under certain circumstances ($U_h$, $T_i$) the near-wake extends beyond 5D, therefore in some farms, the near-wake will influence other turbines.

- Recent work has been conducted to understand the influence of objects within the near-wake of a turbine, for example transmission lines (challenging the current view and guidelines that overhead transmission lines should not be installed within 3D of a wind turbine; see [1]) and bird clusters [5].

We thank the reviewer for their comment on the clarity of combining work by [3] and [2]. The method is now stated more clearly in the manuscript, alongside a modified definition of the novelty of the proposed method.

The manuscript has been updated to reflect the astute comments from the reviewer, shown below, in blue.

The first dataset is a near-wake high-resolution planar PIV wind tunnel experiment at a working TSR of $\lambda = 5.5$. The necessity to accurately predict the near-wake comes not only from the possible interaction with downstream turbines, but, recently, the interaction with other objects within a farm, such as transmission lines or bird clusters ([1] and [5]). As a purely near-wake focus would be limiting, two additional datasets are included in this investigation.

...

An additional model, developed with the combination of the double-Guassian formulation by [3] coupled with the yaw model from [2] is proposed.

...

an additional model, developed with the combination of the double-Guassian formulation by [3] coupled with the yaw model from [2]

...

Following this, Sect. 3 describes the proposed analytical model based on a combination of work by Keane [3] and Bastankhah and Porté-Agel [2].

...

This section proposes a wake model using a combination of existing models, developed to improve the accuracy of predictions in the near-wake under yaw misalignment.

...

Any novel modifications to the equations are highlighted in bold and described in the section below.

...

where $D_{\pm}$ is modified from work by [3] with the addition of the yaw centre $y_c$, and multiplication of $r_{min}$ by $\cos(\gamma)^2$, given by

...

The equation for the wake centre $y_c$ is dependant on the normalised length of the potential core $x_0/D$ given below from work by Bastankhah and Porté-Agel [2]

...

Work by Keane [3] uses the actuator disc model. Keane employs work by Tennekes and Lumley [4] to show the mean momentum flux across a disk is given by

...

This study compared six analytical yaw wake models, including a proposed double-Gaussian formulation as a combination of the work by Keane [3] and Bastankhah and Porté-Agel [2]

Major comments:

1) Reproducibility of the wake data comparison / Missing information on input data.

A quantitative model comparison of models with tunable input parameters is always somehow "tricky". Dependent on the model tuning the difference to the measured reference data can be actively influenced. Here, the authors present a consistent way of model tuning by minimizing the absolute mean

error of every model to the measured reference data. However, it is not transparent to which values the tunable parameters of each model have been set. For reproducibility of the presented results, the values of these parameters is required. I would suggest including an additional table:

Table 2: Analytical wake models – tuning parameters Include all the values of the tunable parameters of the different wake models for the Near-wake / Far-wake / Full-scale case for reproducibility of the presented results.

Futhermore, an overview of the main parameters of the three reference datasets is missing. Most of this information can be found in the cited original papers, while information about the turbine thrust coefficient CT is missing completely. Especially turbine's CT and the turbulence intensity in the inflow are regarded as crucial input parameters to the models. It would be interesting to see how similar/different they are for the three reference datasets.

Table 3: Comparison of the experimental datasets Include information about Turbine size, diameter, CP, CT, inflow-TI, inflow-shear ...

Futhermore, an overview of the main parameters of the three reference datasets is missing. Most of this information can be found in the cited original papers, while information about the turbine thrust coefficient CT is missing completely. Especially turbine's CT and the turbulence intensity in the inflow are regarded as crucial input parameters to the models. It would be interesting to see how similar/different they are for the three reference datasets.

We thank the reviewer for their comment regarding the clarity of model calibration and on the reproducibility of the manuscripts data. The authors agree fully with this comment and have included two additional tables (Table 1 and Table 2) within the manuscript's, shown below, in blue.

The process used to tune these parameters is outlined in Sect. 2.2 and the resulting parameters are displayed in Table 1. The results from the tuning procedure are discussed in the following section.

...

The information necessary to reproduce the results from the near-wake, far-wake and full-scale datasets are presented in Table 2. It must be noted that all yawed thrust coefficients are measured for the near- and far-wake datasets, however, for the full-scale dataset only the zero yaw angle thrust coefficient is known, hence the thrust coefficient is estimated using $\cos^2(\gamma)$, shown in Table 2

2) Experimental details/limitations.

TABLE 1: Tuned parameters for each analytical model. $X_0$ represents the models standard calibration.

| Model | Parameter | Near Wake | Far Wake | Full Scale | $X_0$ |
|---|---|---|---|---|---|
| BPA | | | | | |
| | $\alpha^*$ | 0.733 | 2.05 | 4.16 | 2.32 |
| | $k$ | 0.00560 | 0.0292 | 0.0293 | 0.0220 |
| Sh | | | | | |
| | $k_w$ | 0.0103 | 0.0948 | 0.0644 | 0.0834 |
| | $\sigma_0/D$ | 0.283 | 0.223 | 0.323 | 0.235 |
| Bl | | | | | |
| | $c_s$ | 0.109 | 0.175 | 0.179 | 0.195 |
| | $k$ | 0.0344 | 0.0301 | 0.0461 | 0.0270 |
| | $b_f$ | -0.354 | -1.06 | -0.775 | -1.15 |
| Ba | | | | | |
| | $U^*$ | 0.341 | 0.493 | 0.525 | 0.216 |
| | $\alpha$ | 0.629 | 0.248 | 0.695 | 0.600 |
| Pr | | | | | |
| | $\alpha$ | 0.0613 | 0.0338 | 0.990 | - |
| | $n$ | 0.174 | 0.937 | 0.567 | - |
| | $r'_{min}$ | 0.467 | 0.416 | 0.477 | - |
| | $d'_e$ | 1.06 | 0.881 | 1.02 | - |

There are some important details about the wind turbine experiment missing in the paper:

(a) First of all, what was the measured thrust coefficient CT at the design tip speed ratio? This crucial input parameter for the wake models is not mentioned anywhere.

(b) Secondly, were power and thrust of the turbine measured for different tip speed ratios, i.e. CP-TSR and CT-TSR curves? Did the power peak for the tip speed ratio it was designed for? (TSR = 5.7)?

(c) What was the chord-based Reynolds number for your turbine at the inflow wind speed of Uinf = 7.8 m/s? From the information given in Table A1, the chord-based Reynolds number should be around Rec,root = 26000 at the innermost blade element and Rec,tip = 74000 at the blade tip. According to Giguere and Selig (1998), the airfoil was designed for Rec = 250 000, while the lowest Reynolds number the polars were measured for was Rec = 100 000. A Reynolds-number independent performance of the turbine should be shown for these low Re-numbers (i.e. by Reynolds-number independent CP-TSR curves measured at different inflow speeds) or the mismatch at least discussed.

TABLE 2: Flow and turbine characteristics.

| Dataset | Yaw angle [°] | $C_T$ | $U_\infty$ [ms$^{-1}$] | TI [%] | $D$ [m] | $\lambda$ |
|---|---|---|---|---|---|---|
| Near-wake | | | | | | |
| | -30 | 0.542 | 7.8 | 5.3 | 0.5 | 5.5 |
| | -25 | 0.579 | 7.8 | 5.3 | 0.5 | 5.5 |
| | -20 | 0.618 | 7.8 | 5.3 | 0.5 | 5.5 |
| | -15 | 0.632 | 7.8 | 5.3 | 0.5 | 5.5 |
| | -10 | 0.657 | 7.8 | 5.3 | 0.5 | 5.5 |
| | -5 | 0.674 | 7.8 | 5.3 | 0.5 | 5.5 |
| | 0 | 0.682 | 7.8 | 5.3 | 0.5 | 5.5 |
| | 5 | 0.672 | 7.8 | 5.3 | 0.5 | 5.5 |
| | 10 | 0.653 | 7.8 | 5.3 | 0.5 | 5.5 |
| | 15 | 0.628 | 7.8 | 5.3 | 0.5 | 5.5 |
| | 20 | 0.602 | 7.8 | 5.3 | 0.5 | 5.5 |
| | 25 | 0.557 | 7.8 | 5.3 | 0.5 | 5.5 |
| | 30 | 0.508 | 7.8 | 5.3 | 0.5 | 5.5 |
| Far-wake | | | | | | |
| | 0 | 0.820 | 4.88 | 7.5 | 0.15 | 3.8 |
| | 10 | 0.780 | 4.88 | 7.5 | 0.15 | 3.8 |
| | 20 | 0.730 | 4.88 | 7.5 | 0.15 | 3.8 |
| Full-scale | | | | | | |
| | 0 | 0.780 | 8.6 | 12.5 | 131 | 7 |
| | 11 | $0.750 = 0.78\cos(11)^2$ | 8.6 | 12.5 | 131 | 7 |

(d) What was the wind tunnel blockage ratio in your experiment? It seems to be low enough, but could the wake deflection be influenced by the side walls of the wind tunnel? At least a basic discussion of this should be included in the description of the experimental setup.

(e) The measured near-wake distances at x/D=2.7 +- 0.6 are very similar. If near-wake measurements are deemed important for this model comparison, why were not measurements closer to the turbine, i.e. x/D ¡ 2, performed? Please justify that in the paper.

[Figure]

FIGURE 1: Power coefficient vs TSR.

We appreciate the reviewer's careful and insightful feedback on the experimental details/limitations. The authors have commented on the points (a-e) below:

(a) The measured thrust coefficients are shown in Table 2.

(b) Yes the Cp-TSR and Ct-TSR curves were measured however, to ensure the turbine was operating at the TSR and Cp of the comparison Senvion 6M126 turbine, the TSR is not at the peak Cp, shown in Fig. 1. Additionally, the gradient of Cp-TSR at the location of operation is far shallower than at the peak, therefore changes in TSR with $T_i$ did not cause major changes in the turbines power.

(c) A Cp-TSR at different wind speeds has been completed for this blade and it is observed that a 30% change in Re causes a 10% change in Ct and a 20% change in Cp at the design TSR, however this is taken into account when tuning the models with input parameters of U and Ct. A discussion regarding the Reynolds dependency of the turbine is added to the manuscript, shown below, in blue.

(d) The blockage ratio is 4 % and the transverse deflection of the wake is less than 0.5D, hence the distance from the centre of the wake to the far walls of the tunnel at maximum deflection is over 2D. A discussion on these points are reflected in the manuscript, shown below, in blue.

(e) Due to the length of the turbine nacelle, to avoid unwanted reflections when conducting PIV, the closest FOV the authors could achieve was $\approx 2D$. A discussion regarding the experimental limitations are included in the manuscript, shown below, in blue.

As a result of the low Reynolds numbers associated with scale wind turbine experiments, a study into the Reynolds dependency of the blade is conducted. It is observed that a 30 % change in Reynolds number causes a 10 % change in thrust coefficient $C_T$ and a 20 % change in power coefficient $C_p$ at the design TSR.

...

The wind tunnel geometric blockage due to the wind turbine is < 4% and is accounted for in the free-stream velocity. The maximum transverse deflection of the

wake is less than $0.5D$, hence the distance from the centre of the wake to the far walls of the tunnel at maximum deflection is over $2D$.

...

Due to the length of the turbine nacelle, to avoid unwanted reflections when conducting PIV, the FOV was limited to $\approx 2D$ behind the turbine.

> 3) Discussion/conclusions on "model flexibility" vs "model complexity" of additional tuning parameters in the proposed wake model.
>
> Also, a critical discussion about the interplay of model tuning and error reduction should be included in the paper. Section 5 closes with "the model's flexibility resulting from having tunable parameters". Isn't it somehow expectable that the MAE is reduced when additional tuning parameters are introduced? How does that make the model more complex to handle on the other hand? Include some lines of discussing that.
>
> In the abstract/conclusions it is mentioned: "The proposed double-Gaussian model achieves the lowest absolute mean error across all datasets." This is correct but not surprising as the model has the highest number of tunable parameters. Also, the main contributor to the lowest absolute mean error across all datasets mainly stems from the near-wake results, that some of the other models are not designed for. I propose another, more realistic conclusion along the lines of "The proposed double-Gaussian model shows improved modelling of the near-wake".

We thank the reviewer for their comments, expanding on the necessity for additional discussion regarding the flexibility and complexity of the models used. The authors agree that the introduction of additional tuning parameters can lead to a reduction in error, and that this effect must be interpreted with care. However, the improvement of the near-wake prediction of the proposed model is not just a consequence of the increased tuning parameters, but rather the use of a double-Gaussian formulation [3]. A discussion on the improved flexibility with model complexity has been added to the manuscript, shown below, in blue.

The authors agree that the more realistic abstract/conclusion the reviewer has suggested gives a logical understanding of the proposed findings. Please see the additional changes to the manuscript in blue below.

The proposed double-Gaussian model shows improved modelling of the near-wake.

...

The reasoning for the greater accuracy of the methods is a consequence of their more complex representation of the near-wake profile, alongside the greater number of

tuning parameters, thereby allowing the models greater flexibility for differing turbine operating conditions. An unwanted result of the increased tuning parameters of the super- (Bl) and double-Gaussian (Pr) models is an increase in the complexity of the models. When tuning each model, it took an order of magnitude more iterations for the super- (Bl) and double-Gaussian (Pr) wake models to converge to a local minimum, when compared to the single-Gaussian formulations.

...

The double-Gaussian model (Pr) consistently demonstrates the highest accuracy, achieving an average MAE of 2.6 %, akin to the model's more complex representation of the near-wake profile, alongside the greater number of tuning parameters, thereby allowing the model greater flexibility for differing turbine operating conditions.

...

Overall the proposed double-Gaussian model shows improved modelling of the near-wake.

Minor comments:

1) dU/Uinf axis in wake profile plots: Although the figure text describes that the distance between two vertical lines corresponds to dU/Uinf = 0.25 or 0.50, additional dU/Uinf axes on top of the profile plots would be helpful to quantify the differences in the predictions.

We thank the reviewer for their comment and agree it would be helpful to have additional axis, this has been modified in the manuscript, and shown below in Fig. 2,3,4,5,7,8.

2) Figures 5 and 10: Is it necessary to show all distances between x/D=4 and x/D=11 in these plots? A reduction to x/D = 4,7,9,11 (or similar) would show the same main trends and be more manageable for the reader.

The authors agree, and are thankful to the reviewer for pointing out where figure 5 and 10 may be overwhelming, this has been changed within the manuscript, shown below in Fig. 3,7.

3) Analysis of both the Mean Absolute Error (MAE) and Root Mean Square Error (RMSE): It is very important to quantify the deviations of the model predictions from the measured velocity data. However, the MAE and the RMSE show the absolute same trends for all comparisons made in this article. Wouldn't it be enough to focus on one error quantification method?

We appreciate the reviewer's thoughtful and constructive comment. The authors agree that, for the cases presented in this study, both metrics exhibit similar overall trends across the models. Nevertheless, It is of the authors opinion that the MAE and RMSE are both important metrics when representing the error of each model. As the MAE offers a robust measure of the general prediction error and the RMSE shows where larger deviations of particular points are present, both are necessary when model predictions are evaluated against planar PIV and single point velocity data. For these reasons, we have retained both MAE and RMSE in the manuscript.

Side note: Figure 9 includes 8x2 error comparisons, which is a lot of information for little variation in the conclusions to be drawn from the error comparison. In my opinion error comparisons of three yaw angles, e.g. for gamma = 10, 20, 30 degrees, would be sufficient here.

We thank the reviewer for this helpful and astute observation, as such Fig. 6 has been updated within the manuscript.

Technical comments:

p. 1, Title of the paper. I suggest to extend the title of the paper to "Analytical yaw models for wind turbine wakes: a two-dimensional comparison". The original title seems incomplete and will make it more difficult to find the paper in search engines.

The authors thank the reviewer for their comments regarding the title of the manuscript and agree with the proposed change.

p. 3, L.67: The acronym "FOV" is not defined yet. Include the full "field of view (FOV)".

We thank the reviewer for their comment and have corrected the manuscript, shown below in blue.

Field Of View (FOV)

p. 5, L.119: "A non time-resolved dataset of 3000 snapshots taken (...) To ensure the velocity snapshots are synchronous with the turbine measurements". In this text section there are two points that are not clear to me: (1) Why is the dataset "non time-resolved"? (2) Why do the velocity measurements need to be synchronized with the turbine measurements when only mean values are presented?

> We thank the reviewer for their comments. The measurements need to be non-time resolved to ensure the velocity fields are statistically uncorrelated. The comment of the synchronization of the velocity measurements has been omitted.

p. 10, L.192: "3.1 Model procedure". If there is the numbering "3.1", there should also be "3.2".

> We thank the reviewer for their comment and have updated the manuscript accordingly.

**Mathematical formulation**

p. 10-15, chapters 4 and 5: it is sometime difficult to recall what the "super-Gaussian", "double-Gaussian", ... models are. Could it be better to use the acronyms defined in Table 1 in the text, e.g. "super-Gaussian Bl", "Double-Gaussian Pr" model?

> The authors agree it may be difficult to recall the models based on how they are currently presented in the manuscript. Therefore all comments regarding the models within the manuscript have an additional acronym consistent with Table 1.

p. 11, Figure 4, and P.14 Figure 8. The figure text of both figures refers to downstream distances x/D= 1.05, 1.65 and 3. From the figures itself and the exp. setup presented earlier in Figure 2, downstream distances somewhere around x/D = 2.1, 2.7 and 3.3 seem to be more correct?

> The authors apologise for the oversight on this matter and thank the reviewer for their comment. The modifications to Figure 4 and 8 have been made, shown below in blue.

$x/D = 2.1, 2.7$ and $3.3$

p. 14, L.264: "... overpredict the wake deficit." It looks to me like they "overpredict the wake deflection", and accurately predict the wake deficit.

> The authors agree they slightly over-predict the wake deflection, which is mentioned in the subsequent paragraph "a slight overprediction is observed". Additionally, the sentence "At a yaw angle of $20°$, aside from at $x/D = 4$, all yaw models slightly overpredict the wake deficit." has been removed from the manuscript. The manuscript has been modified to make this point clearer shown below, in blue.

[Figure]

FIGURE 2: Spanwise profiles of the normalised streamwise velocity at $x/D = 2.1, 2.7$ and $3.3$, comparing near-wake experimental data with tuned models at a yaw angle of zero degrees. The solid vertical grey lines indicate $U/U_\infty = 0.5$.

[Figure]

FIGURE 3: Spanwise profiles of the normalised streamwise velocity from $x/D = 4$ to 11, comparing far-wake experimental data with tuned models at a yaw angle of zero degrees. The solid vertical grey lines indicate $U/U_\infty = 0.25$.

All other models generally maintain good agreement with the measured wake centre, although a slight overprediction is observed at $20°$.

[Figure]

FIGURE 4: Spanwise profiles of the normalised streamwise velocity at $x/D = 2$, 4 and 6, comparing full-scale experimental data with tuned models at a yaw angle of zero degrees. The solid vertical grey lines indicate $U/U_\infty = 0.25$.

[Figure]

FIGURE 5: Spanwise profiles of normalised streamwise velocity at $x/D = 2.1$, 2.7 and 3.3, comparing near-wake experimental data with tuned models at (Top) $\gamma = 10°$, and (Bottom) $\gamma = 30°$. The solid vertical grey line marks $U/U_\infty = 0.5$.

[Figure]

FIGURE 6: MAE and RMSE, expressed as percentages of the free-stream velocity, for each analytical model compared with near-wake velocity data at yaw angles ranging from $\gamma = -30°$ to $\gamma = 30°$

[Figure]

FIGURE 7: Spanwise profiles of normalised streamwise velocity from $x/D = 4$ to 11, comparing far-wake experimental data with tuned models at (Top) $\gamma = 10°$, and (Bottom) $\gamma = 20°$. The solid vertical grey line marks $U/U_\infty = 0.25$.

[Figure]

FIGURE 8: Spanwise profiles of normalised streamwise velocity at $x/D = 2$, 4 and 6, comparing full-scale experimental data with tuned models at $\gamma = 11°$. The solid vertical grey line marks $U/U_\infty = 0.25$.

**References**

[1] Bs en 50341-2-4:2019 Overhead electrical lines exceeding ac 1 kv – part 2-4: National Normative Aspects (NNA) for the United Kingdom, 2019.

[2] Majid Bastankhah and Fernando Porté-Agel. Experimental and theoretical study of wind turbine wakes in yawed conditions. *Journal of Fluid Mechanics*, 806: 506–541, 2016. .

[3] Aidan Keane. Advancement of an analytical double-gaussian full wind turbine wake model. *Renewable Energy*, 171:687–708, 2021. .

[4] Hendrik Tennekes and John Leask Lumley. *A first course in turbulence*. MIT press, 1972.

[5] Xuefeng Yang, Lifan Zhou, Yi Sui, Zhengru Ren, and Shengli Chen. Effects of wind turbine wakes on bird gliding aerodynamic performance. *Scientific Reports*, 2025. .

---

## Author Comment (AC2)

**Referee:2**

General comments:

This paper presents a novel set of measurements characterising the near wake of a yawed, small-scale wind turbine in a wind tunnel. By combining these measurements with existing literature data, the authors compare several analytical velocity deficit models coupled with yaw models. A model is proposed and evaluated alongside established approaches. Before comparison, all models are calibrated using non-yawed wind turbine wake data, ensuring a consistent methodology across the analysis. The proposed paper is interesting and includes a new dataset for model validation. However, this first submission suffers from limitations that will be detailed later. Also, the potential impact of this study and its innovative character are not clear to me.

We thank the reviewer for providing valuable positive feedback on our manuscript. In the following document we address each of their comments, enhancing the revised version to meet the standards for publication. Any modifications made to the manuscript are highlighted in blue and, when relevant, have been included below to address the reviewer's specific comment.

The authors would like to take a moment to highlight the novelty of this submission, which comes in three forms:

- The inclusion of a high resolution near-wake dataset.

- A comparison of current analytical models using three experimental datasets (including full-scale), where previous literature would have mostly only included one experimental dataset.

- The proposition of an analytical model as the combination of two previous models.

Specific comments:

The analysis is restricted to lateral profiles, which may be misleading. Wake centers are not consistently located near hub height, contrary to the paper's assumptions. Expanding measurements to include 2D y - z planes would provide a more comprehensive understanding of wake behaviour.

We thank the reviewer for the indication on where greater analysis can be conducted and where the authors have made an oversight in their assumptions. The authors agree with the reviewer that work by [3] (Figure 5) shows the wake centre deviates beyond the hub height plane. From work by [3] an assumption is made in their wake model that "The wake centre in (5.1) is assumed to remain at hub height $z_h$ as the vertical displacement of the wake centre is rather small for lower yaw angles (see figure 5)." The assumption in the manuscript has been modified to reflect the reviewers astute comment, shown below, in blue.

The authors agree that additional planes would offer a more comprehensive understanding of the wake behaviour, however, some of the models analysed in this work are strictly two dimensional. Seeing as 2 dimensional models are still used as a starting point within the industry, the comparison is valid.

Work by Bastankhah and Porté-Agel [3] states "the vertical displacement of the wake centre is rather small for lower yaw angles" hence, when implementing the wake centre deflection model from Bastankhah and Porté-Agel [3] there is an assumption that the wake centre $y_c$ remains at hub height.

The study's focus on the near wake is limiting in view of validating models. Extending measurements to cover the full wake would offer deeper insights into wake evolution and model applicability.

We thank the reviewer for their comment and agree that a purely near-wake focus is limiting. Hence the reason for the two additional datasets used within this investigation, both of which cover the far-wake region. To emphasise this point, the manuscript has been updated below, shown in blue.

The first dataset is a near-wake high-resolution planar PIV wind tunnel experiment at a working TSR of $\lambda = 5.5$. The necessity to accurately predict the near-wake comes not only from the possible interaction with downstream turbines, but, recently, the interaction with other objects within a farm, such as transmission lines or bird clusters ([1] and [17]). As a purely near-wake focus would be limiting, two additional datasets are included in this investigation.

The calibration process is unclear and requires significant clarification. Specifically:

– Which parameters are calibrated, and how?

– Do parameters vary with streamwise location, or are predefined functions assumed (e.g., a linear approach for the Gaussian width sigma or another function for the super-Gaussian order n)?

TABLE 1: Tuned parameters for each analytical model. $X_0$ represents the models standard calibration.

| Model | Parameter | Near Wake | Far Wake | Full Scale | $X_0$ |
|---|---|---|---|---|---|
| BPA | | | | | |
| | $\alpha^*$ | 0.733 | 2.05 | 4.16 | 2.32 |
| | $k$ | 0.00560 | 0.0292 | 0.0293 | 0.0220 |
| | | | | | |
| Sh | | | | | |
| | $k_w$ | 0.0103 | 0.0948 | 0.0644 | 0.0834 |
| | $\sigma_0/D$ | 0.283 | 0.223 | 0.323 | 0.235 |
| | | | | | |
| Bl | | | | | |
| | $c_s$ | 0.109 | 0.175 | 0.179 | 0.195 |
| | $k$ | 0.0344 | 0.0301 | 0.0461 | 0.0270 |
| | $b_f$ | -0.354 | -1.06 | -0.775 | -1.15 |
| | | | | | |
| Ba | | | | | |
| | $U^*$ | 0.341 | 0.493 | 0.525 | 0.216 |
| | $\alpha$ | 0.629 | 0.248 | 0.695 | 0.600 |
| | | | | | |
| Pr | | | | | |
| | $\alpha$ | 0.0613 | 0.0338 | 0.990 | - |
| | $n$ | 0.174 | 0.937 | 0.567 | - |
| | $r'_{min}$ | 0.467 | 0.416 | 0.477 | - |
| | $d'_e$ | 1.06 | 0.881 | 1.02 | - |

– If parameters are set for each streamwise location, how are integrations performed with the super-Gaussian model?

We thank the reviewer for their comments regarding the calibration process. The authors agree that more details can be included for the calibration of the models, hence an additional Table. 1 is included within the manuscript, shown below. The authors reached out to Frèdèric Blondel to obtain their super-Gaussian model, hence the implementation of the integration was consistent with the published article (x1=0). Additional investigations were conducted as to how the initial streamwise location affected the integrations, it was concluded that there was no difference in the result of the super-Guassian model with starting locations beyond x1=0, consistent with code by Frèdèric Blondel.

The process used to tune these parameters is outlined in Sect. 2.2 and the resulting parameters are displayed in Table 1. The results from the tuning procedure are discussed in the following section.

Was the presented model implementation validated in their original form, before calibration?

We thank the reviewer for their comment and confirm that the wake centre was validated with results from [3] and the velocity deficit was validated with results from [8] shown in Fig. 1.

The presented results for the super-Gaussian model deviate from those in other studies using the same dataset (e.g., wind tunnel data from Bastankhah & Port´e-Agel). This discrepancy must be addressed, possibly by validating the implementation. Additionally, reporting calibrated values and comparing to accepted model calibration (i.e., https://doi.org/10.5194/wes-8-141-2023 in the present case) would provide some insight. A sensitivity study on the calibration constant could also provide insight.

The authors thank the astute comment made by the reviewer concerning the validity of the implementation of the super-Gausian model. The authors must apologise for an error when implementing the Upper Incomplete Gamma Function converting between C++ and MATLAB. The model was initially validated at $\gamma = 0°$ (as it was wrongfully assumed the deflection code sent from the author had been correctly converted), however under further inspection and correction, the model has now been fully validated under $\gamma = 10°$, $\gamma = 20°$ and $\gamma = 30°$, shown with blue crosses in Fig. 2. These correction have made changes to the deflection of the super-Gaussian model, hence figures 8, 10, 11, 9, 12 and relevant parts of the results section have been modified, shown below, in blue.

In terms of the prediction of the wake centre, all models capture the deflection well, other than the super-Gaussian model (Bl) which overpredicts the wake centre at both yaw angles.

...

Similarly to the near-wake results, the super-Gaussian model (Bl) overpredicts the yawed wake position at $10°$. This overprediction becomes more pronounced at a yaw angle of $20°$. Similarly, the single-Gaussian model (QI) by Qian and Ishihara [11] overpredicts the wake centre at $20°$.

...

In the near-wake, the single-Gaussian model (QI) developed by Qian and Ishihara [11] performs the worst, followed by the super-Gaussian model (Bl) developed by Blondel et al. [5]. The single-Gaussian model (BPA) by Bastankhah and Porté-Agel [3], the vortex sheet model (Ba), and lifting line model (Sh) show similar behaviour, with MAE values between 4.8 % and 5.9 %. The double-Gaussian model (Pr) closely matches the experimental data achieving a MAE of just 2.2 %.

...

For the full-scale case all models display comparable predictive accuracy, with a MAE within $\pm 0.3$ %. When averaging across all datasets, it is evident that the single-Gaussian model (QI) by Qian and Ishihara [11] has the poorest predictive capability, as expected given its lack of tunable parameters. The super-Gaussian (Bl), single-Gaussian model (BPA) by Bastankhah and Porté-Agel [3], lifting line (Sh), and vortex sheet (Ba) methods exhibit similar performance, with MAE values ranging between 3.6 % and 4.6 %.

...

The double-Gaussian model (Pr) performs best at all angles in the near-wake exhibiting MAE values below 2.5 %s. In contrast, the super-Gaussian model (Bl) performs the worst at large yaw angles ($> 10°$), akin to the underprediction of the wake deficit and overprediction of the wake centre. This suggests not only that a more complex wake shape than the single-Gaussian is required, but also an accurate description of the wake centre.

> Missing information to reproduce the results: based on the information provided in the paper, it is not possible to reproduce the test cases. At least the wind turbine thrust coefficients must be provided, together with other relevant flow and wind turbine operation variables.

We thank the reviewer for their valuable insight and agree that there is information lacking to reproduce the results within the paper. To correct this, Table 2 has been inserted into the manuscript to ensure reproducibility and to give a better understanding of the test conditions for the three experimental datasets.

The information necessary to reproduce the results from the near-wake, far-wake and full-scale datasets are presented in Table 2.

> The method used to estimate the yawed thrust coefficient (CT ,$\gamma$ with $\gamma$ the yaw angle) is unclear. Different approaches exist (e.g., CT cos($\gamma$), CT cos2 ($\gamma$), CT cos3 ($\gamma$)), and consistency across models is essential for fair comparison. A clear presentation of all models and their assumptions is needed.

We thank the reviewer for their insightful feedback on estimating yawed thrust coefficient. Aside from the full-scale dataset all thrust coefficients are measured, hence no additional estimation is required. In the case of the full-scale dataset the $\cos^2$ approach is utilised. These changes have been updated in the manuscript, shown below, in blue.

It must be noted that all yawed thrust coefficients are measured for the near- and far-wake datasets, however, for the full-scale dataset only the zero yaw angle thrust

TABLE 2: Flow and turbine characteristics.

| Dataset | Yaw angle [°] | $C_T$ | $U_\infty$ [ms$^{-1}$] | TI [%] | $D$ [m] | $\lambda$ |
|---|---|---|---|---|---|---|
| Near-wake | | | | | | |
| | -30 | 0.542 | 7.8 | 5.3 | 0.5 | 5.5 |
| | -25 | 0.579 | 7.8 | 5.3 | 0.5 | 5.5 |
| | -20 | 0.618 | 7.8 | 5.3 | 0.5 | 5.5 |
| | -15 | 0.632 | 7.8 | 5.3 | 0.5 | 5.5 |
| | -10 | 0.657 | 7.8 | 5.3 | 0.5 | 5.5 |
| | -5 | 0.674 | 7.8 | 5.3 | 0.5 | 5.5 |
| | 0 | 0.682 | 7.8 | 5.3 | 0.5 | 5.5 |
| | 5 | 0.672 | 7.8 | 5.3 | 0.5 | 5.5 |
| | 10 | 0.653 | 7.8 | 5.3 | 0.5 | 5.5 |
| | 15 | 0.628 | 7.8 | 5.3 | 0.5 | 5.5 |
| | 20 | 0.602 | 7.8 | 5.3 | 0.5 | 5.5 |
| | 25 | 0.557 | 7.8 | 5.3 | 0.5 | 5.5 |
| | 30 | 0.508 | 7.8 | 5.3 | 0.5 | 5.5 |
| Far-wake | | | | | | |
| | 0 | 0.820 | 4.88 | 7.5 | 0.15 | 3.8 |
| | 10 | 0.780 | 4.88 | 7.5 | 0.15 | 3.8 |
| | 20 | 0.730 | 4.88 | 7.5 | 0.15 | 3.8 |
| Full-scale | | | | | | |
| | 0 | 0.780 | 8.6 | 12.5 | 131 | 7 |
| | 11 | $0.750 = 0.78 \cos(11)^2$ | 8.6 | 12.5 | 131 | 7 |

coefficient is known, hence the thrust coefficient is estimated using $\cos^2(\gamma)$, shown in Table 2

The "new" approach combines two existing models, which raises questions about its novelty and the space devoted to it in the paper. If something new is introduced here, please mention it clearly.

We thank the reviewer for their comment on the novelty of the proposed method and agree that it is a combination of two methods, which is now stated more clearly in the manuscript. When reviewing what equations are required from (2)-(20), equations 2,3,4,5,6,7,8,9,10,11,16,17,18,19&20 are necessary. We suggest that equations (12)-(14) are also required to understand the main assumptions and formulation of the model by [8]. These changes are now more clearly highlighted in the manuscript, alongside a modified definition of the novelty of the proposed method, shown below, in blue.

An additional model, developed with the combination of the double-Guassian formulation by [8] coupled with the yaw model from [3] is proposed.

...

an additional model, developed with the combination of the double-Guassian formulation by [8] coupled with the yaw model from [3]

...

Following this, Sect. 3 describes the proposed analytical model based on a combination of work by Keane [8] and Bastankhah and Porté-Agel [3].

...

This section proposes a wake model using a combination of existing models, developed to improve the accuracy of predictions in the near-wake under yaw misalignment.

...

Any novel modifications to the equations are highlighted in bold and described in the section below.

...

where $D_{\pm}$ is modified from work by [8] with the addition of the wake centre $y_c$, and multiplication of $r_{min}$ by $\cos^2(\gamma)$, given by

...

The equation for the wake centre $y_c$ is dependant on the normalised length of the potential core $x_0/D$ given below from work by Bastankhah and Porté-Agel [3]

...

Work by Keane [8] uses the actuator disc model. Keane employs work by Tennekes and Lumley [13] to show the mean momentum flux across a disk is given by

...

This study compared six analytical yaw wake models, including a proposed double-Gaussian formulation as a combination of the work by Keane [8] and Bastankhah and Porté-Agel [3]

While mean streamwise velocities and turbulence intensity are compared to ISO standards, further details are required:

– Clear definition of turbulence intensity (TI)

– Assessment of the lateral velocity component

– Uniformity of velocity profiles in the lateral direction

– Comparison of integral length scales to expectations

– Provision of both lateral and vertical profiles

We thank the reviewer for drawing attention to these important points. A clear definition of turbulence intensity and integral length scale has been added to the manuscript, shown below, in blue.

The mean lateral velocity components (V and W) were measured and are $0 \pm 2\%$ of U at all heights. Additionally, the lateral variation in the mean streamwise velocity and streamwise turbulence intensity is less than 2% at hub height, shown in Fig. 3. Finally, the streamwise integral length scale at hub height was determined to be $L_u = 0.31$ m via the integration of the autocorrelation, and confirmed using the central peak frequency method ($L_u = 0.29$ m), a fit to the von Kármán spectrum ($L_u = 0.30$ m), and the zero-frequency spectral intercept method ($L_u = 0.35$ m). These points are reflected in the manuscript, shown below, in blue.

All measurements within this campaign are taken at a hub-height velocity of $U_h = 7.8$ ms$^{-1}$ with a hub height turbulence intensity of $T_i = 5.3$ % and a hub height streamwise integral length scale of $L_u = 0.31$ m. The streamwise turbulence intensity is calculated using local mean velocity at the specified height with the equation shown below

$$T_i = \frac{\sqrt{\overline{u'^2}}}{U}, \tag{1}$$

where $u'$ is the fluctuations in streamwise velocity. The streamwise integral length scale $L_u = U_h T_l$ is determined via the integration of the autocorrelation, and confirmed using the central peak frequency method, a fit to the von Kármán spectrum, and the zero-frequency spectral intercept method

$$T_l = \int_0^\infty \rho_x(\tau) \, dt, \tag{2}$$

where $\tau$ is the time lag and $\rho_x$ is the autocorrelation coefficient defined as

$$\rho_x(\tau) = \frac{R_x(\tau)}{R_x(0)}, \tag{3}$$

where $R_x$ is the autocorrelation function in time.

...

Figure 1 presents the normalised mean velocity $U/U_h$ and streamwise turbulence intensity $T_i$ profiles

...

At all measured heights, the mean spanwise and vertical velocity components are approximately zero, with magnitudes within $\pm 2\,\%$ of the streamwise velocity component (i.e., $V = 0 \pm 0.02U$ and $W = 0 \pm 0.02U$). Additionally, the variation in the streamwise velocity component $U_h$ and streamwise turbulence intensity $T_i$ at hub height with spanwise location ($\pm 0.6D$) is less than 2%.

> Beyond the new dataset, the study's novelty is unclear. Recent advancements in modelling complex wake shapes under yaw and secondary steering are not addressed, limiting the paper's contribution to the field.

We thank the reviewer for their insight. The novelty of this submission comes in three forms: the inclusion of a high resolution near-wake dataset, a comparison of current analytical models using three experimental datasets (including full-scale), previous literature would mainly have at most one experimental dataset, and the proposition of an analytical model (albeit the combination of previous models). A mention of the recent advancements in complex wake shapes and secondary steering have been included within the manuscript, shown below in blue.

The wake of a turbine under yawed conditions is complex (and shown as non-Gaussian), hence the vortex sheet model, proposed by Bastankhah et al. [4] predicts the wake shape by treating the wake edge as a vortex sheet.

...

Recent models, such as work by King et al. [9] and Howland and Dabiri [6] are capable of predicting secondary steering, where the wake of a downstream turbine is influenced by the wake of an upstream turbine.

> Surprisingly, the widely used Jimenez model is omitted from the comparison. Including it would provide a more comprehensive benchmark for the proposed and existing models.

We are grateful to the reviewer for this thoughtful comment on including the classic Jimenez model. It is of the authors opinion that the inclusion of the Jimenez model, although regularly seen in previous literature, detracts from the current manuscript. In preliminary analyses, the Jiménez model exhibited substantially larger errors under yawed conditions than the other models, which led to a compression of the error scale and obscured the relative performance differences among the remaining models. For this reason, and to preserve the readability and interpretability of the comparative figures, the Jiménez model was omitted from the final comparison.

Technical corrections:

1. L.2: The use of "2D" is ambiguous. Please clarify.

> We thank the reviewer for this comment and have modified the manuscript, shown below, in blue.

This study compares six yawed wake models capable of predicting the streamwise velocity component in the hub height plane. The yaw models evaluated include

2. L.3: The claim that a "new" double-Gaussian model is proposed is misleading. The model used is that of Keane et al., coupled with the Bastankhah & Port´e-Agel yaw model. Please clarify what is new in the proposed approach.

> We thank the reviewer for this comment. The authors have modified the manuscript to ensure there is no confusion with the proposed model, shown below, in blue.

An additional model, developed with the combination of the double-Guassian formulation by [8] coupled with the yaw model from [3] is proposed.

3. L.5: The assertion that measurements serve as an undebatable reference overlooks the complexity of full-scale measurements, which are subject to uncontrolled environmental biases. Are these measurements based on neutral atmospheric conditions? A more critical discussion is needed.

> We thank the reviewer for this comment, a more critical discussion is included in the manuscript, shown below, in blue.

However, all models struggle to predict the full-scale dataset under yawed conditions, possibly from uncontrollable environmental biases, but none the less emphasising the necessity for validating models against a wide range of turbine operating conditions.

4. L.11: The term "more" is vague. Please reformulate for precision and provide references to support the claim.

> We thank the reviewer for this comment, additional references have been included in the manuscript, shown below, in blue.

Although less detailed than numerical simulations, recent analytical models are able to accurately predict the entire wake region under large yaw misalignments, thereby increasing their use in the development of farm-wide control algorithms ([2, 16]).

5. L.17: While the model depends on a single tuning parameter ($\sigma$, I assume), $\sigma$ itself is usually assumed to be a linear function with coefficients dependent on turbine operating and environmental conditions. This should be made explicit.

> We thank the reviewer for this comment and have modified the manuscript, shown below, in blue.

With just a single tuning parameter (the wake expansion) determined from the turbine operating and environmental conditions, the model successfully predicts the far-wake behind a host of turbine geometries subjected to different (non-yawed) inflow conditions.

6. L.29: The mention of wind shear in the Ishihara & Qian paper is unclear. Shear is not a parameter in the models, and previous models (e.g., Bastankhah & Port´e-Agel) are compatible with non-uniform vertical streamwise velocity profiles as input. Clarify the intended meaning.

> We thank the reviewer for this comment and have modified the manuscript, shown below, in blue.

In recent years three-dimensional models are developed based on the idea of axisymmetric self-similarity assumptions ([7, 12, 10, 14]). Some of which include the introduction of an incoming Atmospheric Boundary Layer (ABL) and wind shear.

7. L.37: Insert "that" between "setting" and "a wake steering" for grammatical correctness.

> We thank the reviewer for this comment and have modified the manuscript.

8. L.43: The statement "requires no tunable parameters" is misleading, as the model depends on multiple calibration constants. Furthermore, these constants could also be tuned in this study.

> We thank the reviewer for this comment and have modified the manuscript.

9. A discussion on wake deformation (curled-wake effect) and the complex wake shape behind yawed turbines should be included in the introduction.

> We thank the reviewer for this comment and have modified the manuscript, shown below, in blue.

The wake of a turbine under yawed conditions is complex (and shown as non-Gaussian), hence the vortex sheet model, proposed by Bastankhah et al. [4] predicts the wake shape by treating the wake edge as a vortex sheet.

...

Recent models, such as work by King et al. [9] and Howland and Dabiri [6] are capable of predicting secondary steering, where the wake of a downstream turbine is influenced by the wake of an upstream turbine.

10. L.81: please use the international system of units

We thank the reviewer for this comment, the 10′×5′ wind tunnel is the name of the facility here at Imperial College London.

11. L.84: Define turbulence intensity (TI) clearly: is it based on hub-height velocity or local velocity u(z)?

We thank the reviewer for this comment and have modified the manuscript, shown below, in blue.

All measurements within this campaign are taken at a hub-height velocity of $U_h = 7.8 \text{ ms}^{-1}$ with a hub height turbulence intensity of $T_i = 5.3\,\%$ and a hub height streamwise integral length scale of $L_u = 0.31$ m. The streamwise turbulence intensity is calculated using local mean velocity at the specified height with the equation shown below

$$T_i = \frac{\sqrt{\overline{u'^2}}}{U}, \tag{4}$$

where $u'$ is the fluctuations in streamwise velocity. The streamwise integral length scale $L_u = U_h T_l$ is determined via the integration of the autocorrelation, and confirmed using the central peak frequency method, a fit to the von Kármán spectrum, and the zero-frequency spectral intercept method

$$T_l = \int_0^\infty \rho_x(\tau)\, dt, \tag{5}$$

where $\tau$ is the time lag and $\rho_x$ is the autocorrelation coefficient defined as

$$\rho_x(\tau) = \frac{R_x(\tau)}{R_x(0)}, \tag{6}$$

where $R_x$ is the autocorrelation function in time.

12. L.85: Both positive and negative yaw angles were considered. Did you observe any non-symmetric behavior as claimed in some studies (https://doi.org/10.5194/wes-6-1521-2021)? This should be discussed.

> We are grateful to the reviewer for this thoughtful comment. When mirroring the PIV data around the vertical plane perpendicular to the turbines blades, and then taking the absolute error between each symmetric angle. For example, mirroring $\gamma = -30°$ so it is consistent with $\gamma = 30°$ and taking the absolute difference, there is limited non-symmetric behaviour observed in the mean velocity. More specifically, for all angles a maximum error of 5% of $U_h$ is observed with a mean difference between each angle of 1% of $U_h$. A comment has been added to the manuscript, shown below in blue.

To investigate the symmetry of the PIV data, it is found that when mirroring the data around the vertical plane perpendicular to the turbines blades, and then taking the absolute error between each symmetric angle, there is limited non-symmetric behaviour observed in the mean velocity. More specifically, for all angles a maximum error of 5% of $U_h$ is observed with a mean difference between each angle of 1% of $U_h$.

13. Figure 1.: Define TI and include V /Uh, T iv, T iw, and turbulent length scales for completeness.

> We appreciate the reviewer's thoughtful and constructive comment, however it is of the authors opinion to keep Fig. 1 the same, and instead include the relevant information regarding turbulence parameters within the manuscript, as there is little perceived change in the vertical or lateral directions. The changes to the manuscript are shown below, in blue.

At all measured heights, the mean spanwise and vertical velocity components are approximately zero, with magnitudes within $\pm 2$ % of the streamwise velocity component (i.e., $V = 0 \pm 0.02U$ and $W = 0 \pm 0.02U$). Additionally, the variation in the streamwise velocity component $U_h$ and streamwise turbulence intensity $T_i$ at hub height with spanwise location ($\pm 0.6D$) is less than 2%.

14. Section 2.1.2: Please provide thrust coefficients for both unyawed and yawed cases. Is the current controller representative of real-scale turbines, especially regarding yaw misalignment? L.119: Did you verify the statistical convergence of the procedure? L.123 to L.126: Are these filtering operations standard? If so, provide references.

We thank the reviewer for these comments. As mentioned above Table 2 has been inserted into the manuscript. For the current measurements the tip speed ratio was kept constant, which for an upwind turbine is representative of the operating conditions. The PIV data are well converged: convergence to within ±1% of the mean is achieved within 2000 samples, and since 3000 samples are used in the present study, the mean can be considered fully converged within ±1%. The filtering methods are outlined in [15]. The manuscript has been updated below, in blue.

Finally, a min-max filter, which can be interpreted as a binary filter, is then applied (Tropea et al. [15]).

15. What is the main motivation for focusing on near-wake characteristics in these measurements?

We thank the reviewer for drawing attention to this important point. It is of the authors opinion that we require accurate near wake modelling because:
- Under certain circumstances ($U_h$, $T_i$) the near-wake extends beyond 5D, therefore in some farms, the near-wake will influence other turbines.
- Recent work has been conducted to understand the influence of objects within the near-wake of a turbine, for example transmission lines (challenging the current view and guidelines that overhead transmission lines should not be installed within 3D of a wind turbine; see [1]) and bird clusters [17].

The first dataset is a near-wake high-resolution planar PIV wind tunnel experiment at a working TSR of $\lambda = 5.5$. The necessity to accurately predict the near-wake comes not only from the possible interaction with downstream turbines, but, recently, the interaction with other objects within a farm, such as transmission lines or bird clusters ([1] and [17]). As a purely near-wake focus would be limiting, two additional datasets are included in this investigation.

16. Section 2.2: The calibration process is unclear. Specify which parameters are tuned, and whether they are tuned independently for each streamwise location and test case. Explain how integration is performed in the super-Gaussian model if the parameters are not continuous.

We thank the reviewer for their comments regarding the calibration process. The authors agree that more details can be included for the calibration of the models, hence an additional Table. 1 is included within the manuscript.

17. Section 2.2: It would be informative to compare tuned models to their standard calibration, possibly in an appendix.

> We thank the reviewer for this helpful observation, the authors have included a column in Table 1 to reflect the difference between the tuned parameters and their standard calibration.

18. Section 3: The derivation appears to use the double-Gaussian formulation of Keane et al. with the Bastankhah & Port´e-Agel yaw model. If no new models are proposed, the extensive derivation should be justified or condensed. If new elements are introduced, they should be clearly highlighted.

> We thank the reviewer for their comment on the novelty of the proposed method and agree that it is a combination of two methods, which is now stated more clearly in the manuscript, shown below, in blue.

An additional model, developed with the combination of the double-Guassian formulation by [8] coupled with the yaw model from [3] is proposed.

...

an additional model, developed with the combination of the double-Guassian formulation by [8] coupled with the yaw model from [3]

...

Following this, Sect. 3 describes the proposed analytical model based on a combination of work by Keane [8] and Bastankhah and Porté-Agel [3].

...

This section proposes a wake model using a combination of existing models, developed to improve the accuracy of predictions in the near-wake under yaw misalignment.

...

Any novel modifications to the equations are highlighted in bold and described in the section below.

...

where $D_\pm$ is modified from work by [8] with the addition of the wake centre $y_c$, and multiplication of $r_{min}$ by $\cos^2(\gamma)$, given by

...

The equation for the wake centre $y_c$ is dependant on the normalised length of the potential core $x_0/D$ given below from work by Bastankhah and Porté-Agel [3]

...

Work by Keane [8] uses the actuator disc model. Keane employs work by Tennekes and Lumley [13] to show the mean momentum flux across a disk is given by

...

This study compared six analytical yaw wake models, including a proposed double-Gaussian formulation as a combination of the work by Keane [8] and Bastankhah and Porté-Agel [3]

19. Eq.2: Explicitly state that $\gamma$ corresponds to the yaw angle. Ensure all quantities in the derivation are clearly defined.

> We thank the reviewer for this comment and have modified the manuscript, shown below, in blue.

To ensure a consistent and unbiased analysis of the wake models under yaw misalignment, each model's parameters are tuned at $\gamma = 0°$, where $\gamma$ is the yaw angle, with the exception of the model by Qian and Ishihara [11] as no tuning parameters are required.

...

where $r$ is the distance from the centre of the turbine in the spanwise direction.

...

where $x$ is the distance from the centre of the turbine in the streamwise direction and $\epsilon = (d'_e - r'_{min})/6$.

20. L.152: $\sigma$ is not a Gaussian function; please correct this phrasing.

> We thank the reviewer for this astute comment. The authors have modified the manuscript, shown below, in blue.

The single Gaussian cross-section

21. Eq.6: Provide a reference to the appropriate source for this model.

> We thank the reviewer for this comment and have modified the manuscript, shown below, in blue.

where $D_\pm$ is modified from work by [8] with the addition of the wake centre $y_c$, and multiplication of $r_{min}$ by $\cos^2(\gamma)$, given by

22. L.158: Please clearly distinguish between non-yawed and yawed CT throughout the paper.

> We thank the reviewer for this comment and have modified the manuscript.

23. L.166: The claim that the wake center is aligned with hub-height is invalid. Bastankhah & Port´e-Agel (Figure 5) clearly shows otherwise.

> We thank the reviewer for the indication on where the authors have made an oversight in their assumptions. The authors agree with the reviewer that work by [3] (Figure 5) shows the wake centre deviates beyond the hub height plane. From work by [3] an assumption is made in their wake model that "The wake centre in (5.1) is assumed to remain at hub height $z_h$ as the vertical displacement of the wake centre is rather small for lower yaw angles (see figure 5)." The assumption in the manuscript has been modified to reflect the reviewers astute comment, shown below, in blue.

Work by Bastankhah and Porté-Agel [3] states "the vertical displacement of the wake centre is rather small for lower yaw angles" hence, when implementing the wake centre deflection model from Bastankhah and Porté-Agel [3] there is an assumption that the wake centre $y_c$ remains at hub height.

24. Figure 4.: The x-axis should not be the streamwise distance, x/D, but rather x/D + (u-uh)/uh (or similar).

> We thank the reviewer for their comment and agree it would be helpful to have additional axis, this has been modified in the manuscript, and shown below in Fig. 5,6,7,8,10,11.

25. Figure 5. the poor agreement between Ishihara & Qian's model and measurements is surprising. Was the implementation validated against the original paper's test cases?

> We thank the reviewer for their comment and confirm that the wake model from [11] was validated shown in Fig. 4.

26. L.240: the experimental data also contain a test case at 30 degrees. Why was this not considered in the analysis?

> We thank the reviewer for this perceptive observation however in work by [3], specifically Figure 21, the spanwise distribution of the streamwise velocity for 30° are not published, hence were not used within the current investigation.

27. L.253: The use of a square cosine function to modify rmin requires justification. A simple cosine might be more appropriate for a purely trigonometric transformation.

We appreciate the reviewer's insightful comment and agree that as a projection it should be cosine, however the data suggests that the relationship between the distance of the Gaussian peaks decreases with the cosine squared. Hence, is the implementation within the paper.

[Figure]

FIGURE 1: Validation of Present model (red crosses) with results from [3] (Top) at $\gamma = 10°, 20°$ at $x/D = 3, 5, 7, 9, 11$ and results from [8] (Bottom) at $x/D = 0.65, 1.16, 1.68, 2.2, 2.72, 3.23, 3.75, 4.27, 4.78, 5.3$.

[Figure]

FIGURE 2: Validation of Bl model with results from [5] at $\gamma = 10°, 20°, 30°$ for $x/D = 1, 3, 5, 7, 9$. The model from the authors is shown with blue crosses.

[Figure]

FIGURE 3: (Top) Contour plot of the variation in streamwise velocity profile in the lateral direction, (Bottom left) variation in streamwise velocity profile in the lateral direction and (Bottom right) variation in streamwise turbulence intensity profile in the lateral direction.

[Figure]

FIGURE 4: Validation of model in blue crosses with results from [11] at $x/D = 2, 6, 10$ for $\gamma = 8°, 16°$.

[Figure]

FIGURE 5: Spanwise profiles of the normalised streamwise velocity at $x/D = 2.1, 2.7$ and $3.3$, comparing near-wake experimental data with tuned models at a yaw angle of zero degrees. The solid vertical grey lines indicate $U/U_\infty = 0.5$.

[Figure]

FIGURE 6: Spanwise profiles of the normalised streamwise velocity from $x/D = 4$ to 11, comparing far-wake experimental data with tuned models at a yaw angle of zero degrees. The solid vertical grey lines indicate $U/U_\infty = 0.25$.

[Figure]

FIGURE 7: Spanwise profiles of the normalised streamwise velocity at $x/D = 2$, 4 and 6, comparing full-scale experimental data with tuned models at a yaw angle of zero degrees. The solid vertical grey lines indicate $U/U_\infty = 0.25$.

[Figure]

FIGURE 8: Spanwise profiles of normalised streamwise velocity at $x/D = 2.1, 2.7$ and $3.3$, comparing near-wake experimental data with tuned models at (Top) $\gamma = 10°$, and (Bottom) $\gamma = 30°$. The solid vertical grey line marks $U/U_\infty = 0.5$.

[Figure]

FIGURE 9: MAE and RMSE, expressed as percentages of the free-stream velocity, for each analytical model compared with near-wake velocity data at yaw angles ranging from $\gamma = -30°$ to $\gamma = 30°$

[Figure]

FIGURE 10: Spanwise profiles of normalised streamwise velocity from $x/D = 4$ to 11, comparing far-wake experimental data with tuned models at (Top) $\gamma = 10°$, and (Bottom) $\gamma = 20°$. The solid vertical grey line marks $U/U_\infty = 0.25$.

[Figure]

FIGURE 11: Spanwise profiles of normalised streamwise velocity at $x/D = 2$, 4 and 6, comparing full-scale experimental data with tuned models at $\gamma = 11°$. The solid vertical grey line marks $U/U_\infty = 0.25$.

[Figure]

FIGURE 12: MAE and RMSE, expressed as percentages of the free-stream velocity, for each analytical model compared with near-wake ($\gamma = 10°$), far-wake ($\gamma = 10°$), and full-scale ($\gamma = 11°$) velocity data, along with the mean of all three datasets.

**References**

[1] Bs en 50341-2-4:2019 Overhead electrical lines exceeding ac 1 kv – part 2-4: National Normative Aspects (NNA) for the United Kingdom, 2019.

[2] Mojtaba Maali Amiri, Milad Shadman, and Segen F Estefen. A review of physical and numerical modeling techniques for horizontal-axis wind turbine wakes. *Renewable and Sustainable Energy Reviews*, 193:114279, 2024. .

[3] Majid Bastankhah and Fernando Porté-Agel. Experimental and theoretical study of wind turbine wakes in yawed conditions. *Journal of Fluid Mechanics*, 806: 506–541, 2016. .

[4] Majid Bastankhah, Carl R Shapiro, Sina Shamsoddin, Dennice F Gayme, and Charles Meneveau. A vortex sheet based analytical model of the curled wake behind yawed wind turbines. *Journal of Fluid Mechanics*, 933:A2, 2022. .

[5] Frèdèric Blondel, Marie Cathelain, Pierre-Antoine Joulin, and Pauline Bozonnet. An adaptation of the super-gaussian wake model for yawed wind turbines. In *Journal of Physics: Conference Series*, volume 1618, page 062031. IOP Publishing, 2020. .

[6] Michael F Howland and John O Dabiri. Influence of wake model superposition and secondary steering on model-based wake steering control with scada data assimilation. *Energies*, 14(1):52, 2020. .

[7] Takeshi Ishihara and Guo-Wei Qian. A new gaussian-based analytical wake model for wind turbines considering ambient turbulence intensities and thrust coefficient effects. *Journal of Wind Engineering and Industrial Aerodynamics*, 177: 275–292, 2018. .

[8] Aidan Keane. Advancement of an analytical double-gaussian full wind turbine wake model. *Renewable Energy*, 171:687–708, 2021. .

[9] J. King, P. Fleming, R. King, L. A. Martínez-Tossas, C. J. Bay, R. Mudafort, and E. Simley. Control-oriented model for secondary effects of wake steering. *Wind Energy Science*, 6(3):701–714, 2021. . URL https://wes.copernicus.org/articles/6/701/2021/.

[10] Li Li, Zhi Huang, Mingwei Ge, and Qiying Zhang. A novel three-dimensional analytical model of the added streamwise turbulence intensity for wind-turbine wakes. *Energy*, 238:121806, 2022. .

[11] Guo-Wei Qian and Takeshi Ishihara. A new analytical wake model for yawed wind turbines. *Energies*, 11(3):665, 2018. .

[12] Haiying Sun and Hongxing Yang. Study on an innovative three-dimensional wind turbine wake model. *Applied energy*, 226:483–493, 2018. .

[13] Hendrik Tennekes and John Leask Lumley. *A first course in turbulence*. MIT press, 1972.

[14] Linlin Tian, Yilei Song, Pengcheng Xiao, Ning Zhao, Wenzhong Shen, and Chunling Zhu. A new three-dimensional analytical model for wind turbine wake turbulence intensity predictions. *Renewable Energy*, 189:762–776, 2022. .

[15] Cameron Tropea, Alexander L Yarin, John F Foss, et al. *Springer handbook of experimental fluid mechanics*, volume 1. Springer, 2007.

[16] Li Wang, Mi Dong, Jian Yang, Lei Wang, Sifan Chen, Neven Duić, Young Hoon Joo, and Dongran Song. Wind turbine wakes modeling and applications: Past, present, and future. *Ocean engineering*, 309:118508, 2024. .

[17] Xuefeng Yang, Lifan Zhou, Yi Sui, Zhengru Ren, and Shengli Chen. Effects of wind turbine wakes on bird gliding aerodynamic performance. *Scientific Reports*, 2025. .